# A multi-site German validation of the Interoceptive Accuracy Scale and its relation to psychopathological symptom burden

Sebastian Brand [1,8✉], Annelie Claudia Meis [2,8], Markus Roman Tünte[3,4,8], Jennifer Murphy[5], Joshua Pepe Woller[6], Stefanie Maria Jungmann[1], Michael Witthöft [1], Stefanie Hoehl [3], Mathias Weymar [6,7], Christiane Hermann[2] & Carlos Ventura-Bort[6,8]

Altered interoception is thought to be implicated in the development of psychopathology. Recent proposals highlight the need to differentiate between dimensions of interoception to better understand its relation to mental health. Here, we validated a German version of the Interoceptive Accuracy Scale (IAS) and investigated the relationship between IAS scores and clinical outcomes, across seven samples from four research centers ($N = 3462$). The German IAS version was best explained by a one-factor structure that showed acceptable psychometric properties. We replicated previous findings showing a negative association between IAS scores and measures of alexithymia. IAS scores were negatively related to measures of clinical symptomatology (e.g., anxiety, depressive, and somatoform symptoms) and neurotic traits. These findings suggest that the German IAS is a reliable and valid instrument for subjective interoceptive accuracy. Results emphasize the importance of distinguishing between dimensions of interoception to understand its potential modulatory and protective role in psychopathology.

[1] Department of Clinical Psychology, Psychotherapy and Experimental Psychopathology, Johannes Gutenberg-University Mainz, Mainz, Germany. [2] Department of Clinical Psychology and Psychotherapy, Justus-Liebig-University, Giessen, Germany. [3] Department of Developmental and Educational Psychology, Faculty of Psychology, University of Vienna, Vienna, Austria. [4] Vienna Doctoral School Cognition, Behavior and Neuroscience, University of Vienna, Vienna, Austria. [5] Department of Psychology, Royal Holloway University of London, London, UK. [6] Department of Biological Psychology and Affective Science, Faculty of Human Sciences, University of Potsdam, Potsdam, Germany. [7] Faculty of Health Sciences Brandenburg, University of Potsdam, Potsdam, Germany. [8] These authors contributed equally: Sebastian Brand, Annelie Claudia Meis, Markus Roman Tünte, Carlos Ventura-Bort. ✉email: sebbrand@uni-mainz.de

Words and sayings like *gut feeling* (intuition), *pain in the neck* (being bothered by something in life), or *butterflies in the stomach* (being in love) illustrate the relevance of bodily changes for interpreting and categorizing one's own experiences. The ability to perceive and process internal bodily signals, such as heart rate or gastrointestinal changes, is defined as interoception[1]. Increasing evidence has demonstrated that individual differences in interoceptive abilities are related to a variety of psychological processes and psychopathological symptomatology[2–7]. Further, recent theoretical frameworks[8–10] have given rise to alternative approaches to investigate interoception and its related constructs, thus facilitating the development of alternative measures[11–15]. However, the tools to assess interoception do not build upon these recent theoretical frameworks and/or lack proper validation, especially in the German language (for a German adaptation of an interoception questionnaire see for example the Multidimensional Assessment of Interoceptive Awareness[16]), highlighting the need for adapting and validating alternative methodological approaches. To fill this gap and advance our understanding of interoception and its relation to clinical symptomatology, in the current study, we report the psychometric evaluation of the German version of the recently developed Interoceptive Accuracy Scale (IAS)[12].

In recent decades, interoception has gained a special interest in psychophysiological and clinical research[1,17–20]. For instance, altered interoceptive abilities, like difficulties detecting cardiac signals during rest[21] or during states of homeostatic perturbation[15,22], have been found in people suffering from anxiety, affective, eating and autism spectrum disorders, schizophrenia, and substance abuse[15,21,23–28]. Similarly, low performance on heartbeat perception tasks and low scores on self-report measures of interoception have been related to depressive and somatoform symptomatology[4,29–35]. However, recent meta-analytic approaches have provided contradicting evidence on the relationship between cardiac interoception and mental health in general[36] as well as anxiety-related symptoms in particular[37]. Part of this dissonance may be related to the measurements used to assess interoception. Most of the abovementioned studies operationalized interoception either as individual performance on cardiac-related perception tasks or as questionnaire scores, which have low correspondence with each other[38]. Furthermore, although other experimental approaches exist (e.g., tasks of gastrointestinal perception[39], of respiratory perception[40], or the perception of spontaneous fluctuations in electrodermal activity[41]), the relationship between interoceptive abilities and psychopathology has mostly been tested with cardiac-related tasks, limiting the generalizability to other domains[36,38,42]. Developing alternative tools and taxonomies that help homogenize measurements of interoception would thus help improve our understanding of the relationship between interoception and psychopathology.

One of the most prominent frameworks advocates classifying interoceptive abilities in three dimensions based on the measurements used[2,43]: (1) interoceptive accuracy, i.e., one's objective accuracy in detecting internal bodily signals, commonly assessed with performance measures such as the Heartbeat counting task or Whitehead heartbeat detection task[36,44–46], (2) interoceptive sensibility (also labeled as subjective accuracy), i.e., subjective beliefs about one's interoceptive abilities, typically assessed with questionnaires or confidence ratings[12,47–52], and (3) interoceptive awareness, i.e., the metacognitive awareness of interoceptive accuracy. Within interoceptive-related tasks, interoceptive awareness is typically assessed by calculating the correspondence between objective performance (i.e., interoceptive accuracy), and the beliefs about performance (i.e., interoceptive sensibility), with higher correspondence indicating higher interoceptive awareness[43].

Although the dimensional model of interoception has helped to clarify the relationship between interoceptive abilities and psychological and clinical symptoms[2,53,54], current proposals[1,38], supported by recent findings[12], emphasize the need for an extended taxonomy of interoception with a more precise subdivision of interoceptive abilities. One of these theoretical models is the $2 \times 2$ factorial model of interoception[8], which suggests distinguishing not only between measures of interoception (more objective performance tasks vs. more subjective self-assessments) but also between constructs (interoceptive accuracy vs. interoceptive attention).

Within the $2 \times 2$ factorial model, interoceptive accuracy is understood as correctly perceiving the true state of one's body, while interoceptive attention is defined as the degree to which a person attends to or focuses on bodily changes. The authors suggest that interoceptive accuracy can be objectively measured with tasks such as Heartbeat counting or detection tasks[45,46]. On the other hand, objective measures of interoceptive attention may involve experience-sampling methods that assess the extent to which interoceptive signals are the object of attention[55]. To measure the subjective beliefs of interoceptive accuracy and attention, recently two self-report measures have been developed, the Interoceptive Accuracy Scale (IAS)[12] and the Interoceptive Attention Scale (IATS)[49]. The IAS assesses subjective beliefs about one's ability to accurately perceive interoceptive signals, while the IATS assesses subjective beliefs concerning one's attention to interoceptive sensations.

Providing initial support for the $2 \times 2$ factorial model, recent studies have shown that subjective interoceptive attention and accuracy are differentially related to external criteria, as is the case with alexithymia, a condition characterized by difficulties identifying and describing one's emotions[17]. Whereas subjective interoceptive attention scores have shown none[56] or a positive association with alexithymia scores[57,58], objective and subjective scores on interoceptive accuracy have been negatively associated with alexithymic traits[12,47,56,59], indicating that subjective interoceptive accuracy (i.e., a precise representation of physiological changes) and attention (i.e., a heightened attentiveness towards physiological changes), are independent traits with seemingly opposing associations with self-reported psychological traits[5].

In line with the relationship between alexithymia and poor interoceptive accuracy, recent proposals stress that low interoceptive accuracy might be related to psychopathological symptom burden and vulnerable tendencies, including somatoform, anxiety, depressive symptoms, and trait neuroticism. Previous studies suggest that people with somatic symptomatology (e.g., somatization disorder) tend to misinterpret physical changes as disease signs, which may be indicative of low interoceptive accuracy[35,60,61]. In addition, individuals at risk for anxiety and depression show difficulties in accurately processing their bodily signals and generating appropriate adaptive responses to the environment[62]. Furthermore, neuroticism, as a trait reflecting emotional instability and playing a crucial role in the development of mental illness[63] has been recently associated with poor interoceptive processing[64].

To sum up, recent findings have shown that interoceptive accuracy and attention, when subjectively assessed, are differentially related to external criteria. This emphasizes the need to create and adapt measures that specifically tap into these constructs. Further, this raises the question of whether interoceptive accuracy and attention are related to other clinical symptoms. Bridging these gaps, the present study aimed to create a German version of the IAS and investigate the relationship between

subjective interoceptive accuracy and measures of psycho-pathology, including trait neuroticism and alexithymia as well as, somatoform, anxiety, and depressive symptomatology.

In line with the original[12] and subsequent validations of the IAS[56], in the German version of the questionnaire, we expected to find acceptable psychometric properties (i.e., good internal consistency and test-retest reliability according to current guidelines[65]) as well as significant correlations with other subjective and objective measures of interoception. Replicating previous findings, we expected to observe a negative relation between subjective interoceptive accuracy and alexithymic traits[12]. Furthermore, extending prior results, we expected to observe a negative relationship between the IAS and measures of psychopathology. That is, we expected subjective interoceptive accuracy to be negatively related to somatoform, anxiety, and depressive symptomatology as well as neurotic traits.

## Method
**General information**. Data were collected from seven different samples across four independent centers: Johannes Gutenberg-University of Mainz, University of Vienna, University of Potsdam, and the Justus-Liebig-University Giessen. Most of the studies were planned independently from each other, partly involving multiple samples and slightly different versions of the questionnaire. It was only later that the authors happened to find out about each other's project through the creator of the original Interoceptive Accuracy Scale (IAS), Jennifer Murphy[12]. Given the notable overlap across projects, but also considering the complementing differences, the authors decided to cooperate to provide an agreed German version of the questionnaire for the scientific community (see Supplementary Notes 2).

**Participants**. A total of $N = 3462$ participants across seven samples from four universities took part in the current study. All participants provided informed consent before participation. Data collection was approved by the Ethics Committee of each psychological institute and/or university. Information about gender identity was obtained through self-report (e.g., *welches Geschlecht haben Sie?*, *welchem Geschlecht fühlen Sie sich zugehörig?*, or *Geschlecht:*) with three response alternatives (*männlich*, *weiblich*, and *divers*) which in German can be understood as a question about biological sex or societal gender. Since these were self-reported answers, we interpreted the information as societal gender. No data on race/ethnicity was collected. Participants were excluded if they did not report a high proficiency German level, were underaged, left items unanswered, and/or responded too fast or slow[66].

**Mainz sample 1**. A total of $N = 506$ participants were recruited via social media (e.g., Facebook groups) as interested volunteers from the German general population (Winter 2019). After exclusion, the first Mainz sample (Mainz S1) consisted of $N = 484$ participants, of which 72.1% reported being women, 26.9% men, and 1% non-binary. The mean age was $M = 27.8$ years (SD = 9.7). This study was not preregistered.

**Mainz sample 2**. From the German general population $N = 1616$, individuals were recruited via social media (e.g., Facebook groups) as interested volunteers (Spring 2020). This sample was also used in another study[67]. After exclusion, the second Mainz sample (Mainz S2) consisted of $N = 1509$ participants, of which 79.5% reported being women, 20.0% men, and 0.5% non-binary. The mean age was $M = 33.3$ years (SD = 13.2). This study was not preregistered.

**Vienna sample 1**. $N = 388$ German-speaking individuals, recruited via the online platform prolific (https://www.prolific.co/), participated in the study (Winter 2021/2022) of which 55.7% reported being women, 42.8% men, and 1.5% non-binary. The mean age was $M = 31.0$ years (SD = 10.9). Participants were compensated with approx. 3.50€. This study was preregistered (https://www.aspredicted.org/e6tr3.pdf, 24 June 2021).

**Vienna sample 2**. Participants ($N = 80$) were students from the University of Vienna. A final sample of $N = 77$ (three were excluded due to technical problems or missing questionnaire data) participated in exchange for course credits or as interested volunteers (between Summer 2021 and Spring 2022). 72.2% reported being women, 26.0% men, and 1.2% non-binary. The mean age was $M = 23.5$ years (SD = 6.5). This study was preregistered (https://www.aspredicted.org/e6tr3.pdf, 24 June 2021).

**Potsdam sample 1**. A total of $N = 267$ students from the University of Potsdam underwent the first study, which was administered via Sona Systems (https://www.sona-systems.com/) in exchange for course credits (between Summer 2020 and Winter 2021). Participants who met the exclusion criteria and/or reported having suffered from a neurological or heart disease ($n = 41$) were removed from the analysis, resulting in a final sample of $N = 226$ participants. 83.2% reported being women, 16.0% men, and 1.0% non-binary. The mean age was $M = 22.8$ years (SD = 4.3). This study was preregistered (https://www.aspredicted.org/e6tr3.pdf, 24 June 2021).

**Potsdam sample 2**. A second battery of questionnaires was administered online via Sona Systems (https://www.sona-systems.com/) to $N = 254$ students from the University of Potsdam in exchange for course credits (between Spring 2021 and Winter 2022). 81.1% reported being women and 18.9% being men. The mean age was $M = 24.5$ years (SD = 6.4). Some participants from Potsdam samples 1 and 2 completed the questionnaires a second time (i.e., to evaluate test–retest reliability). Some of the participants recruited in Vienna and Potsdam were used for validation of other interoception-related questionnaires[58] or to investigate their relation to emotional experience[68]. This study was preregistered (https://www.aspredicted.org/e6tr3.pdf, 24 June 2021).

**Giessen sample**. A total of $N = 522$ German-speaking individuals completed an online survey via Unipark[69] (Fall 2021). Participants were remunerated with a 10€ shopping voucher. Participants were recruited via email lists from the Justus-Liebig-University Giessen, via study requests on social media platforms (e.g., Facebook or Instagram groups), and flyers at local stores. No participants had to be excluded. Participants in the final sample reported 79.5% women, 20.0% men, and 0.5% non-binary gender. The mean age of the sample was $M = 23.4$ (SD = 8.4). This study was not preregistered.

**Questionnaires**. Figure 1 offers an overview of the used measurements in each sample.

### Interoception and related questionnaires
*German versions of the IAS*. Following current guidelines[70], the original version of the IAS was first translated into German by independent researchers. Subsequently, a back-translation was performed by a professional interpreter and/or a native speaker. Although all versions were translated from the English version of the IAS[12] versions slightly differed in terms of the wording used (e.g., more formal vs. more informal, see Supplementary Notes 1 for details). In the original English validation, the IAS showed

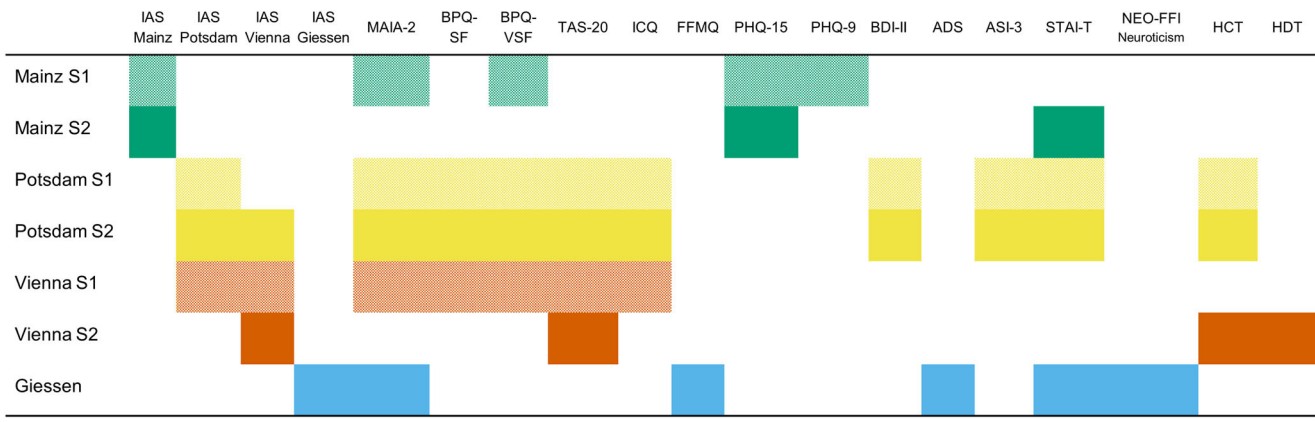

**Fig. 1 Overview of samples and measures.** Note. S1 = Sample 1; S2 = Sample 2; Green boxes mark measurements used in the Mainz samples, Yellow boxes mark measurements used in the Potsdam samples, Orange Boxes mark measurements used in the Vienna samples and blue boxes mark measurements used in the Giessen sample; IAS Interoceptive Accuracy Scale, MAIA-2 Multidimensional Assessment of Interoceptive Awareness Version-2; BPQ-(V)SF Body Perception Questionnaire (Very) Short-Form, TAS-20 Toronto Alexithymia Scale-20, ICQ Interoceptive Confusion Questionnaire, FFMQ Observation and Description subscales of the Five Facet Mindfulness Questionnaire, PHQ-15 Patient Health Questionnaire 15-Item Version, PHQ-9 Patient Health Questionnaire 9-Item Version, ADS German Version of the Center for Epidemiological Studies Depression Scale, BDI-II Beck Depression Inventory, ASI-3 Anxiety Sensitivity Inventory 3, STAI-T State-Trait-Anxiety Inventory Trait-Version, NEO-FFI Neuroticism Neo Five-Factor Inventory Neuroticism subscale, HCT heartbeat counting task, HDT heartbeat detection task.

good psychometrical properties, for example, good to excellent internal consistency[12], $0.88 < \alpha < 0.90$, and good test-retest reliability[12], $r(115) = 0.75$, $p < 0.001$. Furthermore, the authors provided evidence for convergent and divergent validity of the construct of subjective interoceptive accuracy.

*Multidimensional Assessment of Interoceptive Awareness, Version-2 (MAIA-2).* The MAIA-2[50] (German validation[16]) focuses on the evaluation of multiple dimensions of interoception throughout its 37 items divided into 8 scales. The 8 subscales are Noticing (4 items), Non distracting (6 items), Not worrying (5 items), Attention regulation (7 items), Emotional awareness (5 items), Self regulation (4 items), Body listening (3 items), and Trusting (3 items). Each item is rated on a 6-point Likert scale. Overall, in the current samples, the subscales of the MAIA-2 showed an acceptable to good internal consistency: $\omega_{\text{Noticing}} \geq 0.63$, $\omega_{\text{Non distracting}} \geq 0.81$, $\omega_{\text{Not worrying}} \geq 0.72$, $\omega_{\text{Attention regulation}} \geq 0.78$, $\omega_{\text{Emotional awareness}} \geq 0.79$, $\omega_{\text{Self regulation}} \geq 0.78$, $\omega_{\text{Body listening}} \geq 0.76$, and $\omega_{\text{Trusting}} \geq 0.80$.

*Body Perception Questionnaire Short Form (BPQ-SF) and Very Short Form (BPQ-VSF).* The BPQ-SF[51,71] (non-validated German translation) is composed of 46 items scored on a 5-point Likert scale, grouped in three subscales that measure two distinct constructs. The Body Awareness subscale (26 items) quantifies the proportion of time a person reports being aware of sensations in their body. The remaining two subscales, supradiaphragmatic reactivity (i.e., the autonomically-innervated response of organs above the diaphragm, 15 items) and subdiaphragmatic reactivity (i.e., the autonomically-innervated gastrointestinal organs, 6 items; one item referring to the feeling of likeliness to vomit is used in both the supra- and subdiaphragmatic scales), assess the construct of subjectively perceived autonomic nervous system reactivity related to difficulties in the coordination of bodily functions as well as symptoms of stress and autonomic dysregulation. In the current study, the BPQ-SF showed good internal consistency ($\omega_{\text{BodyAwareness}} = 0.92$; $\omega_{\text{Supradiaphragmatic}} = 0.87$; $\omega_{\text{Subdiaphragmatic}} = 0.81$).

In some of our samples (see Fig. 1), the BPQ-VSF[51,71] (non-validated German translation), which comprises 12 items from the body awareness subscale of the BPQ-SF, was administered. To have more comparable measures across samples, the scores from BPQ-VSF were extracted from the BPQ-SF. BPQ-VSF scores showed good internal consistency ($\omega \geq 0.87$). Some items of the existing German version of the BPQ-VSF and BPQ-SF that appeared oddly phrased, were reworded (the translation procedure of such items was similar to the translation of the IAS).

*Interoceptive Confusion Questionnaire (ICQ).* The ICQ[47] (no German validation) consists of 20 items, evaluating the difficulties interpreting one's non-affective physiological states, such as hunger or muscle pain. The ICQ is scored on a 5-point Likert scale. In our studies, the internal consistency of the ICQ was acceptable ($\omega \geq 0.66$). The German translation of the ICQ followed a similar procedure to the IAS translation.

*The Observation and Description subscales of the Five Facet Mindfulness Questionnaire (FFMQ).* The Observation subscale of the FFMQ[72] (German validation[73]) assesses sensory awareness, including how the internal and external world is perceived. The Description scale of the FFMQ evaluates how personal experiences are labeled. Both subscales comprise eight items in a forced-choice answer format, ranging from "never or very rarely true" (1) to "very often or always true" (5). Both subscales showed good internal consistency ($\omega_{\text{Observation}} = 0.75$, $\omega_{\text{Description}} = 0.91$).

**Psychopathology and related questionnaires**

*Toronto Alexithymia Scale (TAS-20).* Alexithymia traits were assessed with the TAS-20[74] (German validation[75]), which consists of 20 items rated on a 5-point forced-choice answer format, grouped in three subscales: Difficulty Identifying Feelings (7 items), Difficulty Describing Feelings (5 items), and Externally Oriented Thinking (8 items). In the current study, the TAS-20 showed good internal consistency ($\omega \geq 0.85$).

*Patient Health Questionnaire 15-Item and 9-Item Version (PHQ-15 and PHQ-9).* Somatic symptom distress was measured using the PHQ-15 (German validation[76]) and depressive symptoms were assessed by the PHQ-9 (German validation[77]). Both Questionnaires are part of a German screening procedure for the assessment of psychological complaints in individuals (PHQ-D)[78]. The PHQ-15 consists of 15 items assessing the degree of individual

somatic symptoms (e.g., abdominal pain) on a 3-point scale. The PHQ-9 consists of 9 items, measuring the degree of individual distress caused by depressive symptoms. The respondents indicate to what extent they are burdened by symptoms such as dejection or hopelessness on a 4-point scale. The PHQ-15 and PHQ-9 showed acceptable internal consistency (Mainz S1: $\omega_{PHQ-15} = 0.77$, $\omega_{PHQ-9} = 0.88$; Mainz S2: $\omega_{PHQ-15} = 0.78$).

*The Beck Depression Inventory (BDI-II).* The BDI-II[79] (German validation[80]) measures the severity of depressive symptoms. It consists of 21 groups of statements assessing the presence of psychological (e.g., feelings of guilt) and physiological (e.g., loss of energy) symptoms of major depression. Statements are assigned point values (ranging from 0 to 3) reflecting the severity of depressive symptoms. In the current sample, the BDI-II showed good internal consistency ($\omega = 0.91$).

*German version of the Center for Epidemiological Studies Depression Scale (ADS).* The ADS (German validation[81]) consists of 20 items, assessing how often depressive symptomatology has been experienced in the last week. Items are rated in a forced-choice format, ranging from "rarely or not at all (less than one day)" (0) to "mostly, all the time (five to seven days long)" (3). In the current sample, the ADS showed good internal consistency, $\omega = 0.92$.

*Anxiety Sensitivity Inventory 3 (ASI-3).* The ASI-3[82] (German validation[83]) measures anxiety sensitivity, a construct referring to a person's fear of their physiological anxiety-related arousal response. The ASI-3 consists of 18 items rated on a 5-point Likert scale. In the current sample, the ASI-3 showed good internal consistency ($\omega = 0.88$).

*State-Trait-Anxiety Inventory (STAI).* The STAI[84] (German validation[85]) intends to measure both state and trait anxiety, in the current study only the trait subscale (STAI-T) was used. This subscale consists of 20 items rated on a 4-point Likert scale. In the present samples, the STAI-T showed good internal consistency ($\omega \geq 0.93$).

*Neo Five-Factor Inventory (NEO-FFI): neuroticism subscale.* Neuroticism was measured using the German version[86] of the neuroticism scale from NEO-FFI[87]. This scale measures the general tendency to experience negative feelings such as fear, sadness, or disgust in 12 statements in a forced-choice answer format, ranging from "not agree at all" (1) to "totally agree" (5). The neuroticism subscale of the NEO-FFI showed good internal consistency ($\omega = 0.87$).

## Objective tasks

*Heartbeat counting task (HCT) and heartbeat detection task (HDT).* Participants from the Potsdam samples ($N = 46$) and the Vienna Sample 2 ($N = 80$) performed a heartbeat counting task (HCT)[46]. In the HCT, participants were instructed to silently count their heartbeats over varying periods without actively touching any body part in which heartbeats could be felt and without trying to guess their heartbeats[88]. To ensure that the interoceptive accuracy scores extracted from the HCT did not reflect any counting strategy (e.g., estimation of the heartbeats based on the time passed) a control, time estimation task was administered (second counting task (SCT))[59,88]. In the SCT, participants are instructed to count the seconds that pass in a specific time interval. An acoustic signal indicated the beginning and end of each trial. After hearing the tone signaling the end of a trial, participants indicated the number of heartbeats felt or seconds counted, and how confident they were

about their response (four trials in the Potsdam samples [25, 35, 45, 100 s or 28, 38, 48, 103 s]; three trials in the Vienna sample [35, 45, 105 s or 38, 48, 103 s]; trials were presented randomly within blocks and trial length was counterbalanced across participants).

Participants in the Vienna Sample 2 also completed a heartbeat detection task (HDT)[45] in which they were presented with a series of 10 tones and had to indicate whether the presentation was synchronous or asynchronous with their heartbeat. There were 40 trials (20 synchronous and 20 asynchronous) presented in randomized order. In synchronous trials, the tones were presented 250 ms after the R-peak, while in asynchronous trials the tones were presented 550 ms after the R-peak.

During HCT, SCT, and HDT, electrocardiography was continuously monitored using a MP-160 BIOPAC system two-lead setup (BIOPAC systems Goleta, California) for the Potsdam sample and an ADInstruments Powerlab 4/35 and Bioamp for the Vienna sample. For the HDT, to time the presentation with the participants' heartbeat, a pulse was sent to a presentation computer using a built-in function of the ADInstruments Powerlab 4/35 (FastResponseOutput) and a custom-built Arduino as an interface between the presentation computer and ADInstruments Powerlab.

For the HCT, we computed the number of heartbeats per participant during each trial by first running scripts performing automatic R-peak detection (Potsdam samples: custom Matlab scripts; Vienna sample: custom Python scripts using Neurokit2) and then visually by inspecting each trial. Trials were rejected if not all R-peaks could be correctly identified e.g., due to movement artifacts. For the HDT, all trials were visually inspected using a custom javascript dashboard, and trials were rejected where not all R-peaks were correctly identified during the stimulus presentation, e.g., due to movement artifacts. For the HDT, two participants were excluded due to a large number of artifacts in the electrocardiographic signal.

From the HCT, different indices of interoception were extracted[43], including interoceptive accuracy (IAcc), interoceptive sensibility, and interoceptive awareness. IAcc was derived from the subjective, counted heartbeats and compared to the objectively measured heartbeats: The accuracy score was calculated for each participant and trial. Interoceptive sensibility scores were derived from the confidence rating about the counted heartbeats (from 0% to 100%). Interoceptive awareness was defined as the absolute difference between the IAcc score and sensibility score in each trial. Similar scores were calculated for the control task, SCT. To ensure normalization of the data, scores were log-transformed IAcc scores and averaged across trials and for each task and participant separately. For the HDT, we computed the percentage of correct responses as a measure of interoceptive accuracy[45].

*Statistical analysis.* The software IBM SPSS Statistics[89], Mplus[90], and R version 4.0.5[91] were used to perform the statistical analyses. Within R, we used the packages *tidyverse*[92], *psych*[93], *lavaan*[94], *lme4*[95], and *cocor*[96].

In the following, we will report data on the individual versions of the questionnaires. Because we did not preregister the intention to combine the samples from Mainz, we first assessed whether they were demographically similar and showed comparable IAS scores. Given that samples differed in age, $t(1991) = 8.52$, $p < 0.001$, $d = 0.45$, 95% CI [0.34, 0.55], gender, $\chi^2(2) = 12.49$, $p = 0.002$, and IAS total scores, $t(1991) = 5.90$, $p < 0.001$, $d = 0.31$, 95% CI [0.21, 0.41], analysis on the Mainz version is reported for each sample, separately. For Vienna and Potsdam versions, in line with the preregistered analysis plan, we pooled together the data across samples (e.g., the Potsdam version was filled out by participants in Vienna sample 1 and Potsdam sample 1 and sample 2, see Fig. 1; see also Supplementary

Notes 9). Data distribution was assumed to be normal, but this was not formally tested.

*Exploratory and confirmatory analysis of the structure of the IAS.* First, an exploratory analysis of the factor structure was performed using a parallel analysis[97]. We decided to perform the exploratory analysis with the second Mainz sample because it contained the largest number of participants. Results from the *parallel criterion* were evaluated but also further criteria such as *Kaiser's criterion* or *scree-criterion* were looked at to determine the most plausible factor solution. Following the most plausible factor solution of the exploratory analysis, we ran a confirmatory analysis on the first Mainz sample, as well as Potsdam, Vienna, and Giessen versions of the IAS, separately.

Confirmatory factor analysis was conducted using the robust mean and variance-adjusted weighted least squares (WLSMV) procedure[98]. The model fit of the exploratory found factor solution was assessed by $\chi^2$-tests. Because of the sensitivity of the $\chi^2$-value to large samples, additional characteristics such as the Root Mean Square Error of Approximation (RMSEA), the Comparative Fit Index (CFI), the Tucker Lewis Index (TLI), and the Standardized Root Mean Square Residual (SRMR) are reported and evaluated according to current guidelines[99,100].

*Descriptive characteristics.* Descriptive characteristics, including mean, standard deviation, skewness, and kurtosis are reported for each version of the IAS (see Supplementary Notes 3 for descriptive characteristics for each item of the versions). Furthermore, to investigate the effects of age on the IAS scores, correlational analysis was performed. Gender differences in IAS scores were assessed using unpaired *t*-tests.

*Internal consistency and test–retest reliability.* Internal consistency was calculated using McDonald's Omega[101]. Test-retest reliability was performed in a subset of participants from the Potsdam samples for both the Potsdam and Vienna version of the IAS. To mimic previous studies on the validation of the IAS[12], test–retest indexes of the IAS and BPQ-SF are reported in the results sections. After completion of the online session, participants could freely sign up for a retest. No initial time limit was imposed between the initial and retest sessions. However, we restricted the analysis to those participants who performed the retest 200 days or less after the initial session. Test–retest reliability was examined, using Pearson's and Spearman's correlation indexes and the Intraclass correlation coefficient (ICC).

Given the wide range of days passed between test and retest (up to 200 days), we examined whether the number of days between test and retest moderated the test-retest relationship. To do so, we performed multiple regressions, using the scores at time 1 (i.e., test) as criterion. The scores at time 2 (i.e., retest), the time passed between time 1 and time 2 (in days) as well as the interaction between both were used as predictor.

*Convergent validity with interoceptive-related and other questionnaires.* To test the convergent validity of the IAS, the relationship between the IAS and other interoceptive scales, including MAIA-2 subscales, BPQ-(V)SF subscales, ICQ, and observation and description subscales of the FFMQ, was examined. The relationship with interoceptive related and other questionnaires was calculated using two-tailed Pearson's correlations. To correct for multiple comparisons (i.e., 13 different self-report measurements), we adapted the significance criterion, using Bonferroni correction ($\alpha = 0.05/13 = 0.003$).

*Relation to objective measures of interoception.* To investigate the relationship between the IAS and objective measures of interoception, we computed two-tailed Pearson's correlations between all relevant indices (i.e., IAcc, interoceptive sensibility, interoceptive awareness, and percentage of correct responses in the HDT) and the IAS total scores.

*Openness and transparency.* The studies conducted in Potsdam and Vienna, which are reported on in this project, have been preregistered at https://www.aspredicted.org/e6tr3.pdf (24 June 2021). During the data collection, it was found that other validation studies were being carried out at the same time. To ensure a comprehensive and uniform validation of the questionnaire, it was decided to collaborate with these other researchers. As a result, the analysis plan had to be changed to create a coherent validation process, which led to a deviation from the original, preregistered analysis plan.

## Results

**Demographic data.** For all versions, we found no evidence for a significant difference in the total score of the IAS between men and women. Furthermore, for the Mainz, Potsdam, and Giessen version, we found a positive significant relationship between age and the IAS total score, $0.08 < rs < 0.24$, $ps < 0.05$ (for details see Table 1, for a histogram of the age distribution across samples and versions, see Supplementary Notes 4).

**Table 1 Descriptive characteristics of IAS sum scores.**

| Samples | Mainz S1 (confirmatory) | Mainz S2 (exploratory) | Potsdam | Vienna | Giessen |
|---|---|---|---|---|---|
| *M* (SD) | | | | | |
| Total | 79.9 (9.3) | 83.1 (10.5) | 79.4 (11.0) | 77.9 (10.3) | 66.1 (9.5) |
| Women | 79.8 (9.5) | 83.0 (10.3) | 79.2 (11.1) | 77.6 (10.2) | 66.0 (9.5) |
| Men | 80.2 (9.0) | 83.8 (10.3) | 79.9 (10.9) | 78.5 (10.6) | 66.7 (9.5) |
| Gender | | | | | |
| *t* (df) | 0.43 (477) | 1.23 (1500) | 0.59 (800) | 1.19 (574) | 0.62 (516) |
| *p* | 0.666 | 0.220 | 0.555 | 0.235 | 0.979 |
| *d* (95% CI) | 0.04 | 0.08 | 0.04 | 0.10 | 0.07 |
| | (0.16, 0.25) | (0.05, 0.21) | (−0.10, 0.19) | (−0.06, 0.25) | (−0.07, 0.13) |
| Age | | | | | |
| *r* (95% CI) | 0.24 | 0.21 | 0.08 | 0.05 | 0.11 |
| | (0.16, 0.32) | (0.16, 0.25) | (0.01, 0.15) | (−0.03, 0.13) | (0.02, 0.19) |
| *p* | <0.001 | <0.001 | 0.025 | 0.223 | 0.014 |
| Skewness (SE) | 0.13 (0.11) | −0.39 (0.06) | −0.36 (0.09) | −0.12 (0.10) | −0.36 (0.11) |
| Kurtosis (SE) | 0.07 (0.22) | 1.33 (0.13) | 1.15 (0.17) | 0.37 (0.19) | −0.21 (0.21) |

*S1* Sample 1, *S2* Sample 2.

**Exploratory factor analysis**. A principal axis factor analysis was conducted on the 21 items. The Kaiser–Meyer–Olkin statistic for calculating sample suitability[102], KMO = 0.92, indicated that the sample was suitable for factor extraction. The parallel analysis showed an ambiguous result with the parallel criterion revealing four factors. Additionally, we found four factors with eigenvalues over *Kaiser's criterion* of 1. However, there was a large drop in eigenvalue after the first factor, resulting in a clear inflexion within the scree plot (eigenvalue first factor: 6.83, eigenvalue second factor: 1.32; *scree-criterium*—see also Supplementary Notes 8). Also, the remaining factors beyond the inflexion showed only a marginal increase in eigenvalue compared to a randomly generated sample (e.g., the 95% percentile of the randomly generated eigenvalue of the second factor was 1.25). Accordingly, we consider the one-factor structure the most plausible one. Factor loadings are shown in Fig. 2 and Table 2.

**Confirmatory testing of the model**. Given the one-factor solution extracted from the exploratory analysis, we fitted the data to a one-factor model, using confirmatory factor analysis. The results of the one-factor solution are shown in Table 3.

Confirmatory factor analysis revealed an acceptable fit of the one-factor solution, as indicated by the SRMR scores[99] ranging from 0.062 to 0.092 (for details see Table 3).

**Internal consistency and test–retest reliability**. All versions of the IAS showed a good internal consistency, $\omega_{Mainz1} = 0.84$, $\omega_{Mainz2} = 0.89$, $\omega_{Vienna} = 0.86$, $\omega_{Potsdam} = 0.88$, and $\omega_{Giessen} = 0.86$.

A total of $N = 115$ participants from Potsdam samples 1 and 2 completed the Potsdam version of the IAS a second time, whereas $N = 57$ participants did so for the Vienna version of the IAS. Replicating previous findings, we saw moderate test-retest reliability in the Potsdam, $r(113) = 0.71$, $p < 0.001$, 95% CI [0.60, 0.79], $r_s (113) = 0.67$, $p < 0.001$, 95% CI [0.55, 0.76], ICC = 0.71, $p < 0.001$, 95% CI [0.60, 0.79], and Vienna version of

the questionnaire, $r(55) = 0.66$, $p < 0.001$, 95% CI [0.53, 0.81], $r_s (55) = 0.59$, $p < 0.001$, 95% CI [0.39, 0.77], ICC = 0.69, $p < 0.001$, 95% CI [0.52, 0.80]. Similar to previous studies on the validation of the IAS[12], test–retest reliability was calculated for the Body Awareness scale of the BPQ-SF, which in contrast to Murphy and colleagues results[12], $r(115) = 0.68$, $p < 0.001$, showed poor test–retest reliability, $r(55) = 0.47$, $p < 0.001$, 95% CI [0.24, 0.65], $r_s (55) = 0.45$, $p < 0.001$, 95% CI [0.21, 0.63], ICC = 0.46, $p < 0.001$, 95% CI [0.23, 0.64].

Test–retest reliability was compared between questionnaires, using Fischer's $Z$ on Pearson's correlations. Results revealed lower test–retest reliability for the body awareness subscale of the BPQ-SF compared to the Potsdam version, $\Delta_r = 0.24$, $Z = -2.27$, $p = 0.023$, Zou's[103] 95% CI [0.03, 0.49]. A similar tendency was observed for the Vienna version, $\Delta_r = 0.23$, $Z = -1.81$, $p = 0.070$, Zou's[103] 95% CI [−0.19, 0.49].

Multiple regressions were performed to test the effects of time passed on the relation between test and retest scores. For the Potsdam IAS version, the scores at time 2 significantly predicted the scores at time 1, $\beta = 0.74$, $t = 6.96$, $p < 0.001$, 95% CI [0.53, 0.96], but no evidence for an effect of days passed, $\beta = 0.02$, $t = 0.20$, $p = 0.841$, 95% CI [−0.18, 0.23], or interaction, $\beta = -0.01$, $t = -0.40$, $p = 0.700$, 95% CI [−0.01, 0.01], was found. Similar results were observed for the Vienna IAS version, as indicated by a significant effect of IAS scores at time 2, $\beta = 0.71$, $t = 4.40$, $p < 0.001$, 95% CI [0.39, 1.04], but no evidence was found for effects of days passed, $\beta = -0.04$, $t = -0.32$, $p = 0.751$, 95% CI [−0.30, 0.22], or the interaction, $\beta = 0.01$, $t = 0.27$, $p = 0.800$, 95% CI [−0.01, 0.01].

**Convergent validity with interoception-related and psychopathological questionnaires**. The complete list of correlations as well as additional intercorrelations between measurements is provided in Supplementary Notes 5.

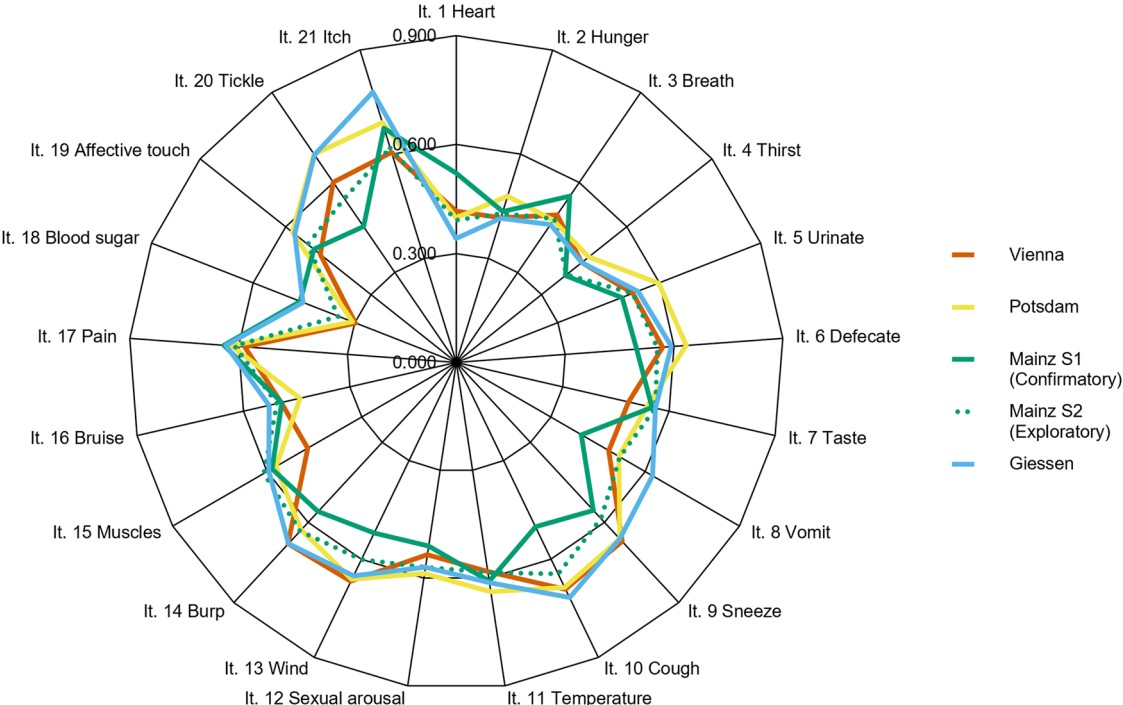

**Fig. 2 Factor loadings of the one-factor solution for each IAS version.** Note. It = Item; The orange line refers to the factor loadings of the Vienna version, the yellow line to the factor loadings of the Potsdam version, the blue line to the factor loadings of the Giessen version and the green line to those of the Mainz version; the green dashed line marks the factor loadings obtained from the exploratory tests on the second Mainz sample.

**Table 2 Factor loadings of the one-factor solution for each IAS version.**

| Items | Factor loadings | | | | |
|---|---|---|---|---|---|
| | Mainz S1 (confirmatory) | Mainz S2 (exploratory) | Potsdam | Vienna | Giessen |
| It. 1 Heart | 0.521 | 0.392 | 0.399 | 0.417 | 0.341 |
| It. 2 Hunger | 0.435 | 0.429 | 0.479 | 0.417 | 0.415 |
| It. 3 Breath | 0.555 | 0.479 | 0.474 | 0.492 | 0.459 |
| It. 4 Thirst | 0.384 | 0.388 | 0.466 | 0.442 | 0.442 |
| It. 5 Urinate | 0.490 | 0.521 | 0.598 | 0.521 | 0.537 |
| It. 6 Defecate | 0.497 | 0.554 | 0.634 | 0.569 | 0.593 |
| It. 7 Taste | 0.553 | 0.562 | 0.539 | 0.486 | 0.562 |
| It. 8 Vomit | 0.397 | 0.519 | 0.516 | 0.484 | 0.623 |
| It. 9 Sneeze | 0.554 | 0.586 | 0.662 | 0.671 | 0.662 |
| It. 10 Cough | 0.502 | 0.646 | 0.686 | 0.692 | 0.718 |
| It. 11 Temperature | 0.608 | 0.584 | 0.637 | 0.582 | 0.612 |
| It. 12 Sexual arousal | 0.510 | 0.569 | 0.587 | 0.534 | 0.569 |
| It. 13 Wind | 0.521 | 0.602 | 0.662 | 0.668 | 0.652 |
| It. 14 Burp | 0.559 | 0.632 | 0.628 | 0.680 | 0.679 |
| It. 15 Muscles | 0.585 | 0.610 | 0.573 | 0.471 | 0.596 |
| It. 16 Bruise | 0.492 | 0.506 | 0.440 | 0.493 | 0.528 |
| It. 17 Pain | 0.640 | 0.611 | 0.619 | 0.584 | 0.635 |
| It. 18 Blood sugar | 0.461 | 0.349 | 0.306 | 0.298 | 0.453 |
| It. 19 Affective touch | 0.500 | 0.522 | 0.576 | 0.478 | 0.568 |
| It. 20 Tickle | 0.453 | 0.546 | 0.694 | 0.601 | 0.692 |
| It. 21 Itch | 0.675 | 0.618 | 0.691 | 0.605 | 0.778 |

*It.* Item, *S1* Sample 1, *S2* Sample 2.

**Table 3 Summary of the indices of model fit of the one-factor solution for each of the versions of the IAS.**

| Model and version | $\chi^2$ | df | p value | RMSEA | CFI | TLI | SRMR |
|---|---|---|---|---|---|---|---|
| Mainz S1 | 572.16 | 189 | <0.001 | 0.065 (90% CI [0.059, 0.071]) | 0.900 | 0.889 | 0.062 |
| Potsdam | 2234.50 | 189 | <0.001 | 0.116 (90% CI [0.111, 0.120]) | 0.775 | 0.750 | 0.082 |
| Vienna | 1962.22 | 189 | <0.001 | 0.121 (90% CI [0.116, 0.126]) | 0.737 | 0.707 | 0.092 |
| Giessen | 939.25 | 189 | <0.001 | 0.087 (90% CI [0.082, 0.093]) | 0.879 | 0.865 | 0.079 |

*S1* Sample 1, *RMSEA* root mean square error of approximation, *CFI* Comparative Fit Index, *TLI* Tucker–Lewis Index, *SRMR* standardized root mean square residual.

**Relationship between the IAS and the MAIA-2 subscales.** The correlation between the different IAS Versions and the MAIA-2 subscales is shown in Fig. 3 (see Supplementary Notes 6 for a detailed description of the correlations and a comparison of correlations across versions). The IAS was positively and moderately correlated with the *Noticing, Attention regulation, Emotional awareness, Self regulation, Body listening,* and *Trusting* subscales. However, we found no evidence for a significant relationship between the IAS scores and the *Non distracting* and *Not worrying* subscales. The pattern of correlations was similar across all questionnaire versions (see Fig. 3).

**Relationship between the IAS and the BPQ-(V)SF**
*BPQ-SF.* For the BPQ-SF, a small positive correlation was found between the IAS and the body awareness subscale, Vienna, $r(640) = 0.32$, $p < 0.001$, 95% CI [0.25, 0.39], Potsdam, $r(806) = 0.31$, $p < 0.001$, 95% CI [0.25, 0.37]. However, after adjusting for multiple comparisons, a small negative association between the supradiaphragmatic and the IAS Vienna version, $r(640) = -0.13$, $p = 0.001$, 95% CI [−0.20, −0.05], and Potsdam version, $r(806) = -0.20$, $p < 0.001$, 95% CI [−0.27, −0.14]), but no evidence for a relationship between the subdiaphragmatic subscale and the IAS Vienna version, $r(640) = -0.08$, $p = 0.041$, 95% CI [−0.16, −0.01], and Potsdam version, $r(806) = -0.10$, $p = 0.004$, 95% CI [−0.17, −0.03]) was found.

*BPQ-VSF.* Overall, a small-to-moderate positive correlation between the IAS versions of Mainz, $r(482) = 0.44$, $p < 0.001$, 95% CI [0.37, 0.51], Potsdam, $r(806) = 0.31$, $p < 0.001$, 95% CI [0.25, 0.37], as well as Vienna, $r(640) = 0.32$, $p < 0.001$, 95% CI [0.25, 0.39], and the BPQ-VSF was found.

*Relationship between the IAS and the TAS-20.* Between the IAS and the TAS-20 a small, negative correlation was found for both the Potsdam, $r(612) = -0.30$, $p < 0.001$, 95% CI [−0.37, −0.22], and the Vienna version, $r(446) = -0.29$, $p < 0.001$, 95% CI [−0.38, −0.21].

*Relationship between the IAS and the ICQ.* The IAS showed a moderate negative correlation with the ICQ for both, the Potsdam, $r(612) = -0.44$, $p < 0.001$, 95% CI [−0.50, −0.37], and Vienna version, $r(446) = -0.50$, $p < 0.001$, 95% CI [−0.57, −0.43].

*Relationship between the IAS and the FFMQ.* Small, significant positive correlations for both the observation, $r(520) = 0.27$, $p < 0.001$, 95% CI [0.19, 0.35], and the description subscales, $r(520) = 0.31$, $p < 0.001$, 95% CI [0.23, 0.39], of the FFMQ with the IAS Giessen version was observed.

*Relationship between the IAS and somatic symptoms (PHQ-15).* A small, negative correlation between subjective interoceptive

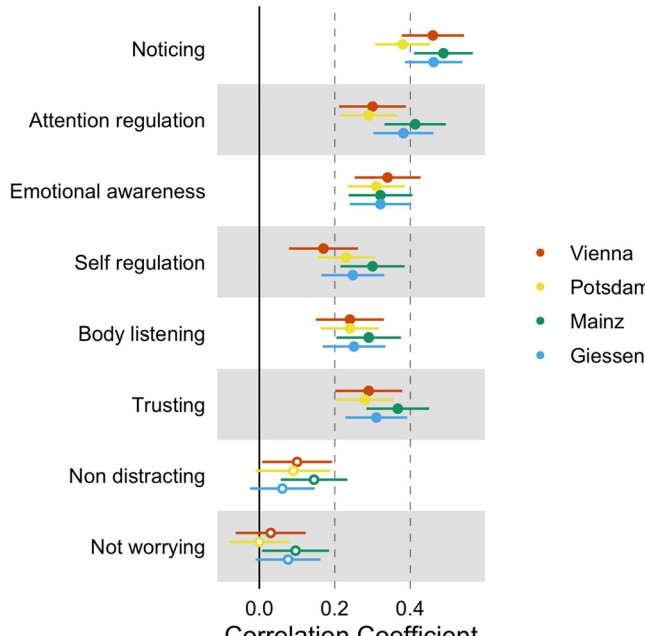

**Fig. 3 Correlations between IAS and MAIA-2 subscales.** Note. The orange dots refer to the correlations between the MAIA-2 subscales and the Vienna version, the yellow dots to the correlations with the Potsdam version, the blue dots to the correlations with the Giessen version and the green dots to the correlation with the Mainz version; filled circles indicate two-tailed, Bonferroni-corrected significant correlations at $p < 0.003$; Error bars represent 95% CIs of Pearson correlation coefficients.

accuracy with somatic symptoms was found in the first, $r(482) = -0.15$, $p < 0.001$, 95% CI [$-0.24$, $-0.07$], and second Mainz sample, $r(1507) = -0.14$, $p < 0.001$, 95% CI [$-0.19$, $-0.09$].

*Relationship between the IAS and depressive symptoms (ADS, BDI-II, and PHQ-9).* A small, negative correlation between subjective interoceptive accuracy and depressive symptoms was found for the ADS, $r(520) = -0.21$, $p < 0.001$, 95% CI [$-0.29$, $-0.12$], as well as for the PHQ-9, $r(442) = -0.19$, $p < 0.001$, 95% CI [$-0.27$, $-0.10$]. Similarly, a significant negative relationship between subjective interoceptive accuracy and depressive symptoms measured via the BDI-II was observed for the Potsdam version, $r(224) = -0.27$, $p < 0.001$, 95% CI [$-0.39$, $-0.15$], but no evidence for a significant relationship was found for the Vienna version, $r(58) = -0.30$, $p = 0.022$, 95% CI [$-0.51$, $-0.05$].

*Relationship between the IAS and anxiety (STAI-T and ASI-3).* Trait anxiety scores were negatively related to subjective interoceptive accuracy across versions, Potsdam, $r(224) = -0.25$, $p < 0.001$, 95% CI [$-0.36$, $-0.12$], Vienna, $r(58) = -0.30$, $p < 0.001$, 95% CI [$-0.51$, $-0.05$], Mainz, $r(1507) = -0.24$, $p < 0.001$, 95% CI [$-0.29$, $-0.19$], Giessen, $r(520) = -0.26$, $p < 0.001$, 95% CI [$-0.34$, $-0.17$]. However, no evidence for significant associations was found between ASI-3 scores and the Potsdam version, $r(224) = -0.15$, $p = 0.025$, 95% CI [$-0.27$, $-0.02$], as well as the Vienna version, $r(58) = -0.06$, $p = 0.652$, 95% CI [$-0.31$, $0.20$].

*Relationship between the IAS and Neuroticism (NEO-FFI).* We found a significant negative correlation between subjective interoceptive accuracy and neuroticism in the Giessen sample, $r(520) = -0.19$, $p < 0.001$, 95% CI [$-0.27$, $-0.11$].

*Relation to objective interoceptive indexes.* Correlations between the IAS Vienna and objective measures of interoception can be found in Table 4. Part of the Potsdam samples ($N = 41$) also completed the IAS Potsdam version and the HCT. Due to the small sample size, we chose to only report correlations with the IAS Vienna version in the main text. For correlations between the IAS Potsdam version and HCT see Supplementary Notes 7. Under the $\alpha = 0.05$ threshold, the IAS was significantly positively correlated with Interoceptive Sensibility (i.e., confidence ratings of the HCT), $r(103) = 0.21$, $p = 0.033$, 95% CI [$0.02$, $0.38$], but no evidence for a relationship with other measures of objective interoception was found. Further, we did not find evidence for a significant relationship between either index of HCT and accuracy on the HDT (further details and control analysis adjusting for potential confounders are listed in Supplementary Notes 7 and 10).

## Discussion

Given the need to provide validated tools that build on recent models of interoception, as well as a need to understand the association between interoceptive constructs and psychopathology, the current study ($N = 3462$, from seven different samples across four research centers) aimed to (1) validate the German version of the recently developed Interoceptive Accuracy Scale (IAS)[12], and (2) investigate its association with symptoms of psychopathology, including depressive, anxious, somatic, alexithymic symptomatology, as well as neurotic traits. The German version of the IAS showed similar psychometric properties to the original English version of the questionnaire[12]. Moreover, self-reported interoceptive accuracy was consistently and negatively related to several clinical psychological measures. These findings support existing models of interoception, highlighting the importance of the construct of subjective interoceptive accuracy to improve our understanding of the relation between interoception and psychopathology[8].

In four independently created (albeit similar) German versions of the IAS, we observed that a one-factor structure could fit the data acceptably. The one-factor structure is in line with the rationale of the original construction of the IAS[12] and the findings of subsequent validations[56]. The German versions of the IAS showed good internal consistency and moderate test-retest reliability. More interestingly, and in line with previous findings[12,56], the German version of the IAS showed higher test-retest reliability than the body awareness scale of the BPQ, suggesting that the self-reported accuracy might be a more stable construct than self-reported awareness.

Regarding construct validity, in line with our expectations, the German version of the IAS was consistently and positively related to other subjective measures of interoception. Consequently, self-reported accuracy was negatively related to measures assessing difficulties in perceiving and understanding bodily signals. Unlike previous findings, showing none[12,49] or a quadratic relationship between IAS scores and the body awareness scale of the BPQ[56], in the current study, we consistently observed a positive relationship between both scales. One possible explanation for the disparity of results across studies might be related to the interpretation of the word 'awareness' in the body awareness scale. It has recently been observed that the relation between the body awareness scale of the BPQ-(V)SF and self-reported interoceptive accuracy and attention is dependent on participants' interpretation. Participants interpreting the scale as a measure of attention showed a stronger relationship with subjective measures of interoceptive attention than those who interpreted the scale as assessing accuracy, whereas the opposite was true for the relationship with the measures of subjective interoceptive accuracy[49].

**Table 4 Correlations between the IAS Vienna and measures derived from the heartbeat counting task (N = 105) and the heartbeat detection task (N = 75).**

| Variable | M | SD | 1 | 2 | 3 | 4 |
|---|---|---|---|---|---|---|
| 1. IAS | 76.5 | 8.3 | | | | |
| 2. HCT | 3.0 | 1.2 | 0.15 | | | |
| 3. HCT interoceptive sensibility | 58.2 | 21.5 | 0.21* | −0.21* | | |
| 4. HCT interoceptive awareness | 35.4 | 25.4 | 0.01 | −0.70*** | 0.55*** | |
| 5. HDT accuracy | 0.4 | 0.1 | 0.12 | −0.02 | −0.14 | −0.01 |

HCT accuracy scores are log-transformed scores derived from counted and actual heartbeats; HCT sensibility refers to the confidence ratings given after each trial and interoceptive awareness is computed from HCT accuracy and sensibility scores. HDT accuracy refers to the percentage of correct responses.
*IAS* Interoceptive Accuracy Scale, *HCT* heartbeat counting task, *HDT* heartbeat detection task.
*$p < 0.05$, ***$p < 0.01$, two-tailed.

Contrary to what was expected, in the present study no evidence for a significant association between objective interoceptive accuracy, as extracted from the HDT and HCT, and IAS scores was found. Previous studies assessing the IAS and objective measures of interoception (HCT) have reported mixed findings with some accounts reporting a significant relationship[12] and others failing to do so[104]. Going beyond previous findings, we also do not find evidence for a relationship with HDT scores. However, subjective interoceptive accuracy as indexed by confidence ratings of the HCT (i.e., interoceptive sensibility) was positively associated with IAS scores. It must also be noted that objective measures of interoceptive accuracy, extracted from the HCT and HDT were unrelated. Although these results were somewhat unexpected, they are not at odds with existing data, as indicated in a recent meta-analysis where only a small association was found between the objective scores of both measures[105]. These findings thus suggest that scores from both tasks may tap into somewhat different aspects of interoception due to differing tasks demands[105]. Although the current findings may provide initial evidence for the construct of subjective interoceptive accuracy, future research on the taxonomy of interoception and the associated objective and subjective correlates is warranted.

Replicating previous results, self-reported interoceptive accuracy was negatively related to alexithymic traits[12,49]. More interestingly, subjective interoceptive accuracy was negatively related to depressive, anxious, and somatic symptoms as well as neuroticism. Our results provide empirical evidence for the existing theoretical models, namely predictive coding models of interoception, emphasizing the role of subjective interoceptive accuracy in the development and symptomatology of associated mental disorders[31,32,34,35,62]. Importantly, these associations could be replicated across samples, measures, and versions of the questionnaire, indicating a stable pattern. The association between self-reported interoceptive accuracy and psychopathological symptom burden suggests that the construct of subjective interoceptive accuracy may be related to a more general factor of psychopathology[106,107] that reflects shared variance across indicators of mental health (i.e., disorders). Further evidence for an association between interoception and psychopathology comes from recent studies showing deficits in cardiac interoceptive accuracy across clinical patients. For instance, researchers observed that, in contrast to healthy control participants, patients suffering from anxiety, depression, eating disorders, and substance abuse have difficulties in improving their interoceptive accuracy in a heartbeat tapping task during an altered physiological state[15,22]. Despite observing a similar negative association between interoception and psychopathology symptoms, the mentioned[22] and our results are based on different measures of interoception which might be unrelated (see above). Future studies should thus focus on identifying the overlapping mechanisms underlying the dimensions of subjective and objective measures of interoception that may predict psychopathological symptom burden.

Although the underlying mechanism relating lower interoceptive accuracy to clinical symptomatology is still unclear, recent proposals embedded within the predictive processing framework may provide valuable insights[15,28]. Predictive processing is a theory of neural functioning and cortical configuration, suggesting that the brain creates generative models of the internal (body) and external (environmental) world to infer the most probable cause of the ongoing changes to efficiently maintain and distribute energetic resources (i.e., allostasis) with the final goal to reduce uncertainty (i.e., free energy)[108]. The generative models try to anticipate resources needed in the upcoming future by making predictions (i.e., generating expectations). These predictions are contrasted with peripheral somatic information (i.e., prediction errors), and updated accordingly, based on the weight (i.e., precision) that the incoming information receives. It is hypothesized that the development and maintenance of mental disorders may be related to a deficient regulation caused by the inability to update the models based on prediction errors, especially in challenging situations[15,22,61,62]. Subjective interoceptive accuracy may thus reflect a general tendency to precisely update generative models based on prediction errors. Future studies investigating the relationship between subjective interoceptive accuracy and computational predictive models of interoception[11,22] may provide more evidence in this regard.

**Limitations.** Despite the strengths of the current findings, some limitations and considerations need to be acknowledged. Our samples consisted of "healthy" young individuals, which might constrain the generalizability of our findings to clinical samples. Future studies investigating the relationship between subjective interoceptive accuracy and clinical symptomatology in patients may help to extend the current results. Related to that, increasing efforts to recruit more heterogeneous samples would further benefit generalizability. It should also be critically mentioned that we made use of translated, but non-validated interoceptive scales (e.g., BPQ and ICQ). Although using non-adapted scales might undermine the current findings, the limited number of German-validated interoceptive questionnaires such as the Multidimensional Assessment of Interoceptive Awareness[16] made it necessary to implement other scales to further test the validity of the IAS. Regarding psychometric goodness in general, it must be noted that only the absolute fit index SRMR showed acceptable goodness of fit across all samples[99]. Although it could be shown that the SRMR is a robust fit index[109] and in previous studies, the goodness of fit of the IAS one-factor solution was similarly imperfect (RMSEA = 0.085, 90% CI [0.077, 0.093], CFI = 0.806, TLI = 0.784)[12], future research should further investigate how the measurement of subjective interoceptive accuracy can be refined

(e.g., critically evaluating the underlying factor structure and/or removing difficult to answer items). For instance, a recent study showed that allowing certain items to correlate (i.e., restrict a model for correlated residuals as a non-theoretical, more data-driven approach) improves the goodness of fit for the one-factor IAS solution (RMSEA = 0.072, 90% CI [0.067, 0.078], CFI = 0.958, TLI = 0.952)[110]. This may indicate that the questionnaire may benefit from grouping different sensations under more overarching terms, as it may be that some sensations are highly correlated and/or hard to perceive[12].

## Conclusion

An interesting, albeit not pre-planned, feature of the present study is the use of 4 different translations of the IAS. Although there were subtle differences in wording (e.g., regarding formality, or the use of different but synonym words of "accuracy" [*genau vs akkurat*]; see Supplementary Notes 1) we found no evidence that these differences impacted the psychometric properties or the relation to symptoms of psychopathology (see Figs. 2 and 3 and Supplementary Notes 6). This indicates that, at least for self-reported interoceptive accuracy as measured by the IAS, such subtle differences might not have a strong impact on the results. Most importantly, joining forces allowed us to provide more compelling evidence for the validation of the IAS and its association with clinical symptomatology (i.e., showing that the relation of subjective interoceptive accuracy to symptoms of psychopathology is replicable and generalizable to different samples). Altogether, our results indicate that the IAS is an acceptable, reliable, and valid instrument for assessing subjective interoceptive accuracy (we recommend the usage of the IAS version from Supplementary Notes 2 in future research). Our findings add further information to the yet very heterogenous empirical evidence on interoceptive abilities and may help understand and refine common theoretical frameworks like the $2 \times 2$ factorial model of interoception. Furthermore, the present study emphasizes the need to distinguish between different constructs of interoception in relation to psychopathological symptom burden.

**Reporting summary**. Further information on research design is available in the Nature Portfolio Reporting Summary linked to this article.

## Data availability

The dataset for the study is available at Open Science Framework[111]. The DOI for this website is https://doi.org/10.17605/OSF.IO/3F2H6. The permanent URL pointing to this raw data is https://osf.io/3f2h6/.

## Code availability

The analysis code for the study is available at Open Science Framework[111]. The DOI for this website is https://doi.org/10.17605/OSF.IO/3F2H6. The permanent URL pointing to this analysis code is https://osf.io/3f2h6/. Study materials are available upon request to the corresponding author without undue reservations.

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

## Acknowledgements
We would like to thank Paula Blum for her support in collecting data for the first Mainz sample as part of her master's thesis, and Laura Neumann for her assistance with the translation of the IAS Vienna version. M.R.T. and S.H. are funded by the FWF (Project number: P33486). Further, M.R.T. was funded by a Förderstipendium of the University of Vienna and S. B. was funded by Verein zur Förderung der Wissenschaftlichen Weiterbildung zum Psychologischen Psychotherapeuten sowie von Forschung und Lehre e.V. The funders had no role in study design, data collection and analysis, the decision to publish, or manuscript preparation.

## Author contributions
S.B. contributed to conceptualization, data curation, formal analysis, investigation, methodology, project administration, resources, validation, visualization, writing—original draft, and writing—review, and editing. A.C.M. contributed to conceptualization, data curation, formal analysis, investigation, methodology, project administration, resources, validation, visualization, writing—original draft, writing—review, and editing. M.R.T. contributed to conceptualization, data curation, formal analysis, investigation, methodology, project administration, resources, validation, visualization, writing—original draft, writing—review, editing, and funding acquisition. J.M. contributed to conceptualization, resources, supervision, and writing—review, and editing. J.P.W. contributed to conceptualization, resources, supervision, and writing—review, and editing. S.M.J. contributed to conceptualization, resources, supervision, and writing—review, and editing. M.W. contributed to conceptualization, resources, supervision, and writing—review, and editing. S.H. contributed to conceptualization, resources, supervision, and writing—review, editing, and funding acquisition. M.W. contributed to conceptualization, resources, supervision, and writing—review, and editing. C.H. contributed to conceptualization, resources, supervision, and writing—review, and editing. C.V.-B. contributed to conceptualization, data curation, formal analysis, investigation, methodology, project administration, resources, validation, visualization, writing—original draft, and writing—review, and editing.

## Funding

## Competing interests
The authors declare no competing interests.
