## [Peer Review File · Communications Psychology]

3rd May 23

Dear Mr Brand,

Thank you for your patience during the peer-review process. Your manuscript titled "Bridging the Gap between Interoception and Mental Health: The German Validation of the Interoceptive Accuracy Scale (IAS) and its Relation to Psychopathological Symptom Burden in a Multicenter Study" has now been seen by 2 reviewers, and I include their comments at the end of this message. They find your work of interest, but raised some important points. We are interested in the possibility of publishing your study in Communications Psychology, but would like to consider your responses to these concerns and assess a revised manuscript before we make a final decision on publication.

We therefore invite you to revise and resubmit your manuscript, along with a point-by-point response to the reviewers. Please highlight all changes in the manuscript text file.

Editorially, we consider it crucially important to fully address the reviewers' concerns regarding the methods and results. While some concerns can be satisfied through changes in presentation, others require further data analysis. In this context, we highlight Reviewer 2's request for transparency regarding the genesis of the project as independent data collections that provide validation for each other. To address this concern, you need to provide alternate analytic strategies as suggested by the referee.

In describing null results, please use appropriate language. (There is no statistical test that can demonstrate absence of an effect. Statements such as 'There is no difference between x and y.' or 'X does not affect Y.' must be revised to read 'We found [no/little] evidence of a difference between x and y.' or 'We found [no/little] credible evidence that X affects Y.'). Please note and implement the comprehensive guidance for statistics reporting and other formatting issues that is offered in the checklist and template linked below.

Please use the following link to submit your revised manuscript, point-by-point response to the referees' comments (which should be in a separate document to any cover letter) and the completed checklist:

[link redacted]

Please do not hesitate to contact me if you have any questions or would like to discuss these revisions further. We look forward to seeing the revised manuscript and thank you for the opportunity to review your work.

Best regards,

Jennifer Bellingtier

Jennifer Bellingtier, PhD
Senior Editor
Communications Psychology

EDITORIAL POLICIES AND FORMATTING

Editorial Policy: [Policy requirements](https://www.nature.com/documents/nr-editorial-policy-checklist.pdf) (Download the link to your computer as a PDF.)

Furthermore, please align your manuscript with our format requirements, which are summarized on the following checklist:

[Communications Psychology formatting checklist](https://www.nature.com/documents/commsj-style-formatting-checklist-review-perspective.pdf)

and also in our style and formatting guide [Communications Psychology formatting guide](https://www.nature.com/documents/commspsychol-style-formatting-guide-accept.pdf) .

*** TRANSPARENT PEER REVIEW:** Communications Psychology uses a transparent peer review system. This means that we publish the editorial decision letters including Reviewers' comments to the authors and the author rebuttal letters online as a supplementary peer review file. However, on author request, confidential information and data can be removed from the published reviewer reports and rebuttal letters prior to publication. If your manuscript has been previously reviewed at another journal, those Reviewers' comments would not form part of the published peer review file.

*** CODE AVAILABILITY:** All Communications Psychology manuscripts must include a section titled "Code Availability" at the end of the methods section. In the event of publication, we require that the custom analysis code supporting your conclusions is made available in a publicly accessible repository; at publication, we ask you to choose a repository that provides a DOI for the code; the link to the repository and the DOI will need to be included in the Code Availability statement.

Publication as Supplementary Information will not suffice. We ask you to prepare code at this stage, to avoid delays later on in the process.

*** DATA AVAILABILITY:**

All Communications Psychology manuscripts must include a section titled "Data Availability" at the end of the Methods section or main text (if no Methods). More information on this policy, is available at <http://www.nature.com/authors/policies/data/data-availability-statements-data-citations.pdf>.

At a minimum the Data availability statement must explain how the data can be obtained and whether there are any restrictions on data sharing. Communications Psychology strongly endorses open sharing of data. If you do make your data openly available, please include in the statement:

We recommend submitting the data to discipline-specific, community-recognized repositories, where possible and a list of recommended repositories is provided at <http://www.nature.com/sdata/policies/repositories>.

If a community resource is unavailable, data can be submitted to generalist repositories such as [figshare](https://figshare.com/) or [Dryad Digital Repository](http://datadryad.org/). Please provide a unique identifier for the data (for example a DOI or a permanent URL) in the data availability statement, if possible. If the repository does not provide identifiers, we encourage authors to supply the search terms that will return the data. For data that have been obtained from publicly available sources, please provide a URL and the specific data product name in the data availability statement. Data with a DOI should be further cited in the methods reference section.

REVIEWERS' EXPERTISE:

Reviewer #1 Interoception, Mental health
Reviewer #2 Mental health, psychometrics

REVIEWERS' COMMENTS:

Reviewer #1 (Remarks to the Author):

I have reviewed the manuscript "Bridging the Gap between Interoception and Mental Health: 2 The German Validation of the Interoceptive Accuracy Scale (IAS) and its Relation to 3 Psychopathological

Symptom Burden in a Multicenter Study”.

The main aim of this study was to validate the German version of the Interoceptive Accuracy Scale (IAS; Murphy et al., 2020) across four research centres in Germany.

Exploratory and Confirmatory analysis were conducted on the Mainz version, and on the Mainz, Potsdam, Vienna, and Giessen versions of the IAS, respectively.

Scores of the slightly different German IAS versions were correlated with commonly used existing interoceptive measures as well as with psychopathological questionnaires, establishing convergent validity with these measures.

The results pointed to a one-factor structure of the IAS with acceptable psychometric properties. Several correlations between the German IAS versions and psychopathological questionnaires replicated previous findings, overall suggesting that the German IAS is a valid and reliable measure of subjective interoceptive accuracy.

The study is convincing and particularly powerful through the combination of data from multiple research centres. Comparisons of measures across the different centres largely demonstrated convergent validity. The similarities and differences between the various German IAS versions were evaluated and effectively used to propose a final (validated) German IAS version that will be useful future research with German speaking participants. I recommend the manuscript for publication after consideration of the following points:

- P.8, L. 187 ff. Participants’ mean ages and SDs have been stated for each sample. Given that three out of four centres found significant positive relationships between the IAS and age, more detail concerning the age distribution would be appropriate. E.g., it appears that overall, the 3 centres showing IAS - Age relationships had higher mean ages and wider age spreads. Please insert either the age range for each sample, or better still, a histogram illustrating the age distribution for each sample (this could be summarised across samples for each centre to match with the results presented in S4 (p.50., L. 1072).
- Correlational analysis throughout the manuscript: please add whether the results are one or two-tailed below the figures / tables and in the data analysis section.
- Several correlational results are based on multiple relationships between the different versions of the IAS and other sub-scales (e.g. BPQ, MAIA, and psychopathological questionnaires). These will need to be corrected for multiple comparisons (e.g. Bonferroni) and clearly stated in the data analysis section. This specifically refers to the results presented in Figure 3 and the correlation matrices in S6.
- P. 15, L. 337. Participants in the Vienna and Potsdam sample carried out the HCT/HDT and the Second Counting task. Please clarify whether and how the Second Counting task has been used as a control measure for correlating IAS with HCT/HDT.
- P. 46, L. 1030. The description of the Giessen translation is a little unclear: do the ‘anchors’ refer to a different formulation of the scale used in the Giessen version, such that Starke Ablehnung = Nie; Ablehnung = Fast Nie; Zustimmung = Fast Immer; Starke Zustimmung = Immer?
A slight reformulation of the description might help to clarify this.
- P. 12, L. 267. Please double-check the mismatch between the total number of items in the BPQ-SF (46) and the individual items stated for each subscale, which amount to 47 (26 + 15 + 6 = 47).

- P. 19, L. 443. Please double-check the reported *r*-value between IAS and Age, which includes a second negative value (or is this intended to show confidence intervals in which case one value seems to be missing)?
- P. 22, L. 481. The test-retest reliability findings for the Body Awareness scale of the BPQ-SF seem to be only reported for the Vienna version (N = 57). What were the Potsdam results?
- General 'cosmetic' changes: pay attention to the consistent use of italics when reporting statistical values (see, e.g. p. 22, L. 481).

Reviewer #3 (Remarks to the Author):

In this submission, the authors report the psychometric properties and the validation of the German translation of the IAS (Interoceptive Accuracy Scale). The manuscript has a very unusual approach. Four different universities in Germany have tested their own slightly different translations on seven different samples. The authors who have learned about the others research only after data collection have decided to publish their findings in one joint publication. They partly merge their data and partly present them in parallel.

This approach raises several issues that need to be addressed.

- The authors identify their submission as a multicenter study. This is inappropriate and also misleading. A multicenter study is characterized by the initial planning to perform the same procedures at different locations, which is not the case here. Rather the submission contains the description of data from four locations and seven samples, i.e. seven independent studies.
- Whether the data presented here can be considered as one large sample asking the same questions and testing the same hypotheses is debatable. If the authors want to present them as one large trial on different locations they need to argue that these data are drawn from the same population. This is clearly not the case. If they are drawn from different populations the authors should test the hypothesis that the samples do not differ. Thus, it is recommended to perform ANOVAs on the samples to test for differences in the variables age, sex, IAS scores, depression scores. Only if there is no difference in these tests data can be merged.
- The other crucial issue are the differences in translations. There are four different translations and the aim of the authors is to validate these translations. The strategy they chose is to present one of the translation as the final one (Mainz) in the supplement. Based on this they should present the deviations of the three other translations from the one chosen. This is presently not the case. In S1 differences for all translations compare to different ones are described. This should be more systematic. Also, it is not clear whether the list in S1 describe *all* differences or represent only a selection. A complete list of all differences need to be presented in S1.
- The differences listed in S1 refer only to one difficult item and seem acceptable. However, in one translation completely different anchor points have been used. This is a decisive difference. Thus, these data should in no case be merged with the other data. The difference should be discussed critically in the Discussion section.
- Rather than considering these data as one large data set from a multicenter study it is suggested to see them as different data sets from different samples and different studies (e.g. with respect to number of additional questionnaires, position of the IAS questionnaire in the whole assessment, study rationale, online vs. in person, before Covid vs. during Covid, different remuneration, different

recruitment strategies, etc.). These studies can then validate (or disconfirm each other) and this should be the main direction of the ms. One idea would be to choose one larger sample as the major one and then to see whether the other samples validate these findings or not.

In the present ms. the data is sometimes merged (e.g. EFA) without justification and sometimes presented separately (e.g. CFA), which is not a systematic approach.

Other issues:

- Line 88: I am not sure whether this statement is true. If something is in the 'focus of attention' then a larger body of research should exist, which is obviously not the case.
- Line 91-92: What are these altered interoceptive abilities? Please give a few examples.
- Line 91ff: There seems to be a general problem with assessing interoceptive abilities. As far as the authors describe there is only one experimental approach to validate subjective reports and that is cardiac interoception. Furthermore, the authors have shown in their data that there is hardly any covariation of this variable with subjective reports. So please give example of other interoceptive tests. If there are none the limitations of this approach should be addressed. I think one cannot conclude from the ability to count ones heartbeats to general interoceptive abilities.
- Line 112-114 Issue (3) is difficult to understand, please explain in a different or more detailed way or give an example.
- Line 140 – 144 Here it would be good to add a sentence with likely interpretations for these findings.
- Line 165-168. It would be helpful to list at one instance in the ms. the psychometric properties of the English IAS. This might be here or in the Results or Discussion section.
- Line 257ff please always mention if scales are validated in German (or not).
- Line 270 -273. This sentence is difficult to understand. Please try to find different wordings and/or explain the constructs (supra- vs. subdiaphragmatic; autonomic dysregulation vs. ANS reactivity).
- Line 337-349ff. It is not clear whether HCT and SCT are two different task or part of the same task. Is the SCT seen as a test of interoceptive ability? Or as a control condition? Please carefully review that paragraph and separate the tests if possible (also with respect to the references).
- Line 418 second word: this should read 'criterion' instead of 'predictor'. Why was the regression done backward in time rather than a forward prediction?
- Line 427 to 436. The description of these variables is not detailed enough to understand/replicate them. It is suggested to move this part to the description of the respective tasks.
- Table 1: The fit indices of the CFA shows an unsatisfying fit in all samples. This challenges the one factor solution and the suitability of the German translation. It is suggested to take this issue up in more detail in the discussion (currently it is only shortly mentioned in limitations). Please also mention if the fit is better or worse than in the English version. How can this result be explained? What are the consequences for the application of the German IAS and for future research? Should the scale be modified or shortened to arrive at a better fit?
- Line 481. It is not clear why the retest reliability of just this test is reported here. Please provide some context information.
- Line 600. I don't share the interpretation that the German word 'Bewusstsein' as used e.g. in 'Körperbewusstsein' (body awareness) is closer to accuracy than attention. Thus, I think this statement is debatable.
- Line 616-618. It is not clear how this lack in covariation "provide(s) support for existing theoretical accounts of interoception...". The finding is obviously contrary to the hypotheses of the theoretical frame the authors use. This warrants a more refined discussion and interpretation, since such results challenge the whole approach of the IAS...
- Line 634 – 636 This interpretation implicitly assumes that subjective introspective accuracy and

objective accuracy are highly correlated/the same. However, the authors' data above have shown that this is not the case. Thus, it cannot be said that these findings are "in line with" Smith et al. 2021, 2022.

- Line 679-680 This statement is in sharp contrast to the earlier interpretation that a slightly different understanding of the word "Bewusstsein" in German will result in substantial differences.
- Line 687 This statement can be debated. In my view there is some support but also some problems (e.g. lack of correlation with objective measures). Also one half of the information with respect to the 2x2 model is missing, since only accuracy is addressed here. Thus, this statement seems to be a bit too bold in the light of the data reported.

We appreciate the time and effort that you and the Reviewers have dedicated to providing such valuable feedback.

As suggested by Reviewer 2, we restructured our analysis plan to align with the genesis of the project. Given that most of the samples were independently collected (and only put together, *post hoc*). We treated them as such unless otherwise specified in our preregistration. For similar reasons, we avoid referring to our study as a “multicenter” one, and thus have changed the title, accordingly. As suggested by Reviewer 1, we have added more information about the age distribution of the samples. The answer to these comments raised by the Reviewers are provided point-by-point here below.

Comments from Reviewer #1:

Comment 1: P.8, L. 187 ff. Participants’ mean ages and SDs have been stated for each sample. Given that three out of four centres found significant positive relationships between the IAS and age, more detail concerning the age distribution would be appropriate. E.g., it appears that overall, the 3 centres showing IAS - Age relationships had higher mean ages and wider age spreads. Please insert either the age range for each sample, or better still, a histogram illustrating the age distribution for each sample (this could be summarised across samples for each centre to match with the results presented in S4 (p.50., L. 1072).

Response: We thank the Reviewer for raising this point. We agree that a more comprehensive description of the age distribution would improve the interpretation of our results therefore, we added a histogram in Supplementary Material S4 to further clarify the results (see also here below).

Age distribution across samples and versions

Comment 2: Correlational analysis throughout the manuscript: please add whether the results are one or two-tailed below the figures / tables and in the data analysis section.

Response: We apologize for not clarifying this in advance. We now state that the correlational

analysis is two-tailed. This information has been provided both in the main text and as a note in the correlation tables and figures, see [Lines 460, 465, 539, 599, 1171, 1177, 1184, 1194]

Comment 3: Several correlational results are based on multiple relationships between the different versions of the IAS and other sub-scales (e.g. BPO, MAIA, and psychopathological questionnaires). These will need to be corrected for multiple comparisons (e.g. Bonferroni) and clearly stated in the data analysis section. This specifically refers to the results presented in Figure 3 and the correlation matrices in S6.

Response: We appreciate the suggestion to correct for multiple comparisons. We agree with Reviewer 1 and thus have applied a Bonferroni correction, lowering the significance threshold from 0.05 to 0.003 [Lines 461-462]:

- *To correct for multiple comparisons (i.e., 13 different self-report measurements), we adapted the significance criterion, using Bonferroni correction ($\alpha = .05 / 13 = .003$).*

With the new corrected threshold, although most of the results remain significant, some of our statements had to be adjusted.

The relation between IAS scores and the non-distracting and non-worrying subscales of the MAIA-2 turned nonsignificant for all of the versions (see Figure 3 and Lines 544-549). In addition, the correlations between the IAS and subdiaphragmatic subscale of the BPO, [Lines 571-574] the BDI-II (for the Vienna version), and ASI-3 [Lines 579-580] scores did not surpass the Bonferroni-corrected significance threshold.

Comment 4: P. 15, L. 337. Participants in the Vienna and Potsdam sample carried out the HCT/HDT and the Second Counting task. Please clarify whether and how the Second Counting task has been used as a control measure for correlating IAS with HCT/HDT.

Response: We thank Reviewer 1 for raising awareness on this and apologize for not clarifying it before. The SCT was utilized as a control condition for the interoceptive accuracy scores extracted from the HCT, rather than as an interoception-related task. We now stated this differentiation more explicitly in the text:

- *To ensure that the interoceptive accuracy scores extracted from the HCT did not reflect any counting strategy (e.g., estimation of the heartbeats based on the time passed) a control, time estimation task was administered (second counting task, SCT; Desmedt et al., 2020; Murphy et al., 2018). In the SCT, participants are instructed to count the seconds that pass in a specific time interval. [365-368]*

We have also deleted the mention of the SCT in the heading of the relevant paragraph to avoid confusion [Line 360].

We further added analysis on interoceptive accuracy scores controlling for scores in the SCT in supplementary material S11; see also answer to comment 14 from Reviewer 2).

Comment 5: P. 46, L. 1030. The description of the Giessen translation is a little unclear: do the 'anchors' refer to a different formulation of the scale used in the Giessen version, such that Starke Ablehnung = Nie; Ablehnung = Fast Nie; Zustimmung = Fast Immer; Starke Zustimmung = Immer? A slight reformulation of the description might help to clarify this.

Response: Thank you for pointing this out. We now rephrased the description to make it clearer. Furthermore, following suggestions from Reviewer 2, differences between IAS versions are highlighted in relation to the final one (Vienna, Potsdam, and Giessen vs Mainz; see supplementary material S1):

- *The initial wording in the Giessen version of the scale was Ich kann genau wahrnehmen which differed from Ich kann immer genau wahrnehmen from the final IAS version. In addition, the wording of the 5-point Likert Scale range from fast nie to fast immer rather than Starke Ablehnung to Starke Zustimmung. [Line 1119-1123]*

Comment 6: P. 12, L. 267. Please double-check the mismatch between the total number of items in the BPQ-SF (46) and the individual items stated for each subscale, which amount to 47 (26 + 15 + 6 = 47).

Response: Thank you for pointing this out. Scales were scored following the suggestions from the original validation of the questionnaire, in which one item (referring to the feeling of likeliness to vomit) is used for both the supra- and subdiaphragmatic scales. We now explicitly state this information in the methods section [Line 289-290]:

- *(...) one item referring to the feeling of likeliness to vomit is used in both the supra-and subdiaphragmatic scales (...)*

Comment 7: P. 19, L. 443. Please double-check the reported r-value between IAS and Age, which includes a second negative value (or is this intended to show confidence intervals in which case one value seems to be missing)?

Response: We intended to provide the correlation range values across IAS versions. To make this more explicit, we now refer to it as follows [Line 472-473]:

- *.08 < rs < .24, p < .05*

Comment 8: P. 22, L. 481. The test-retest reliability findings for the Body Awareness scale of the BPQ-SF seem to be only reported for the Vienna version (N = 57). What were the Potsdam results?

Response: We would like to clarify that we only used one version of the BPQ-SF. Furthermore, the test-retest scores are only provided by the Potsdam sample. To avoid misunderstandings, we now highlight that only the Potsdam sample participated in the re-test [Line 506].

Comment 9: General 'cosmetic' changes: pay attention to the consistent use of italics when reporting statistical values (see, e.g. p. 22, L. 481).

Response: Amended.

Comments from Reviewer #3:

Comment 1: The authors identify their submission as a multicenter study. This is inappropriate and also misleading. A multicenter study is characterized by the initial planning to perform the same procedures at different locations, which is not the case here. Rather the submission contains the description of data from four locations and seven samples, i.e., seven independent studies.

Response: We thank the Reviewer for pointing this out and apologize for the usage of misleading terminology. We agree that the term "multicenter" does not fully apply to the current study given that many of the studies were carried out independently. Therefore, we have decided to omit "multicenter" throughout the manuscript. The amended parts are:

- *Bridging the Gap between Interoception and Mental Health: The German Validation of the Interoceptive Accuracy Scale (IAS) and its Relation to Psychopathological Symptom Burden [Title]*
- *In the current study, we validated the German version of the recently developed Interoceptive Accuracy Scale (IAS; Murphy et al., 2020) and investigated its relation to clinical outcomes, across seven samples from four research centers [Line 50-53]*

- Instead of Keyword “*Multicenter Study*” now “*Validation*” [Line 62]
- *To fill this gap and advance our understanding of interoception and its relation to clinical symptomatology, in the current study, (...)* [Line 80-82]
- *A total of N = 3462 participants across seven samples took place in the current study.* [Line 202]
- *(...) psychopathology, the current study (N = 3462, from seven different samples across four research centers)* [Line 608-609]

Comment 2: Whether the data presented here can be considered as one large sample asking the same questions and testing the same hypotheses is debatable. If the authors want to present them as one large trial on different they need to argue that these data are drawn from the same population. This is clearly not the case. If they are drawn from different populations the authors should test the hypothesis that the samples do not differ. Thus, it is recommended to perform ANOVAs on the samples to test for differences in the variables age, sex, IAS scores, depression scores. Only if there is no difference in these tests data can be merged.

Response: We are very thankful for this suggestion and acknowledge the importance of addressing this issue to ensure the validity and interpretation of our results. It must be noted, however, that in some cases (Vienna and Potsdam samples), joining different samples was part of the initial pre-registered analysis plan. Taking this into consideration, and following Reviewer 2’s suggestion, we proceeded as follows: if collapsing the samples was not part of our pre-registration plan (e.g., Mainz sample 1 and Mainz sample 2), we first tested whether samples differed in terms of age, gender, and IAS scores and only collapsed them if they showed similar results. If collapsing samples was part of the pre-registration plan (i.e., Vienna and Potsdam samples and versions), we combined them. In addition, and for the sake of transparency, we further provided additional analysis on the Vienna and Potsdam samples. Those include comparing them in terms of age, gender, and IAS scores as well as correlational analysis for each sample and IAS version with TAS-20, BPQ-SF MAIA-2 questionnaires (Supplementary material S10). These changes are described in the main manuscript as follows [Lines 412-419]:

- *In the following, we will report data on the individual versions of the questionnaires. Because we did not preregister the intention to combine samples 1 and 2 from Mainz, we first assessed whether they were demographically similar and showed comparable IAS scores. Given that samples differed in age, gender, and IAS total scores, $ps < .001$, analysis on the Mainz version will be reported for each sample, separately. For Vienna and Potsdam versions, in line with our preregistered analysis plan (aspredicted.org/e6tr3.pdf), we pooled together the data across samples (e.g., the Potsdam version was filled out by participants in Vienna sample 1 and Potsdam sample 1 and sample 2, see Figure 1; see also Supplementary Material S10).*

Due to the now split of Mainz samples 1 and 2, the following minor changes are to be mentioned:

- [Line 329-330] reads now “*acceptable internal consistency (Mainz S1: $\omega_{PHQ-15} = .77$, $\omega_{PHQ-9} = .88$; Mainz S2: $\omega_{PHQ-15} = .78$).*”
- [Line 478-481] Minor changes in eigenvalues
- [Line 504] Added internal consistency for each sample
- [Line 566] Added correlation with PHQ-15 for both samples
- Supplementary Material S3 and S4: We now report Descriptive Statistics for each sample
- Splitting up Factor Loadings in S5 and Figure 2

We also deleted the following sentence: *Despite that, given identical conceptual meaning, data from all versions were used indistinctively to refer to the German version of the IAS [Formerly Line 268], to avoid further confusion about merging the data as one large sample.*

Furthermore, Supplementary Material S10 contains the following information: :

- *Following our preregistration (aspredicted.org/e6tr3.pdf), for our main analysis we pooled together the Vienna and Potsdam samples to increase the sample size available for the Vienna and Potsdam translations. Despite our pre-planned analysis strategy, we assessed whether samples differed in age, gender and IAS scores. We observed that the Vienna and Potsdam samples differed regarding both age, $t(572.41) = -11.46, p < .001$, and gender, $\chi^2(2) = 62.19, p < .001$, however, mean scores of the IAS Vienna, $t(567.15) = .59, p = .56$, and Potsdam, $t(805.96) = -.47, p = .64$, versions were not different for the Vienna and Potsdam samples.*

Additionally, we computed correlational analysis for the questionnaires available in both the Vienna and Potsdam samples (MAIA II, BPQ, ICQ, TAS-20). We divided the IAS into samples and versions meaning that we compared 4 versions of the IAS (Vienna sample & Vienna version, Vienna sample & Potsdam version, Potsdam sample & Vienna version, Potsdam sample & Potsdam version). A forestplot with all correlations is depicted here below.

Comment 3: The other crucial issue are the differences in translations. There are four different translations and the aim of the authors is to validate these translations. The strategy they chose is to present one of the translation as the final one (Mainz) in the supplement. Based on this they should present the deviations of the three other translations from the one chosen. This is presently not the case. In S1 differences for all translations compare to different ones are described. This should be more systematic. Also, it is not clear whether the list in S1 describe **all** differences or represent only a selection. A complete list of all differences need to be presented in S1.

Response: Thank you for your valuable suggestion to improve the presentation of S1. We agree that S1 needs a more systematic approach and have therefore clarified in the title that the differences from the final version presented are "all" differences. We have also deleted the paragraph referring

to the Mainz version (as this version is provided in S2) and written down the differences of each other version to the Mainz version.

Comment 4: The differences listed in S1 refer only to one difficult item and seem acceptable. However, in one translation completely different anchor points have been used. This is a decisive difference. Thus, these data should in no case be merged with the other data. The difference should be discussed critically in the Discussion section.

Response: We apologize for any confusion that may have arisen from our description of the differences in the questionnaire versions. We would like to clarify that we have not merged the data from the different questionnaire versions. We acknowledge your observation that completely different anchor points were used in one of the translations. We agree that this discrepancy is a notable difference and should be treated with caution. The revised version of our manuscript contains a more detailed and thorough analysis of the differences between the questionnaire versions (see supplementary material S1 and the previous comment).

Comment 5: Rather than considering these data as one large data set from a multicenter study it is suggested to see them as different data sets from different samples and different studies (e.g. with respect to number of additional questionnaires, position of the IAS questionnaire in the whole assessment, study rationale, online vs. in person, before Covid vs. during Covid, different remuneration, different recruitment strategies, etc.). These studies can then validate (or disconfirm each other) and this should be the main direction of the ms. One idea would be to choose one larger sample as the major one and then to see whether the other samples validate these findings or not. In the present ms. the data is sometimes merged (e.g. EFA) without justification and sometimes presented separately (e.g. CFA), which is not a systematic approach.

Response: We appreciate the Reviewer's suggestions on the analysis plan. As suggested, we conducted the exploratory analysis on our largest data set (Mainz sample 2). Therefore, we have restructured the analysis part of the manuscript as follows:

- *We decided to perform the **exploratory analysis with the Mainz S2** sample because it contained the largest number of participants. Results from the parallel criterion were evaluated but also further criteria such as Kaiser's criterion or scree-criterion were looked at to determine the most plausible factor solution. Following the most plausible factor solution of the exploratory analysis, we ran **a confirmatory analysis on the Mainz S1 sample, as well as Potsdam, Vienna, and Giessen versions of the IAS, separately.** [Line 422-427]*

Comment 6: Line88: I am not sure whether this statement is true. If something is in the 'focus of attention' then a larger body of research should exist, which is obviously not the case.

Response: We apologize for not providing an accurate statement. We changed it, accordingly [Line 84-86]:

- *In recent decades, interoception has gained a special interest in psychophysiological and clinical research (e.g., Brewer et al., 2021; Herbert et al., 2020; Khalsa et al., 2018; Tsakiris & De Preester, 2018; Vaitl, 1996).*

Comment 7: Line 91-92: What are these altered interoceptive abilities? Please give a few examples.

Response: We now provided some examples:

- *For instance, altered interoceptive abilities, like difficulties detecting cardiac signals during rest (e.g., Ardizzi et al., 2016) or during states of homeostatic perturbation (e.g., Smith et al., 2020, 2021), (...) [Line 86-88]*
- *Similarly, low performance on heartbeat perception tasks and low scores on self-report measures of interoception have been related to (...) [Line 92-93]*

Comment 8: Line 91ff: There seems to be a general problem with assessing interoceptive abilities. As far as the authors describe there is only one experimental approach to validate subjective reports and that is cardiac interoception. Furthermore, the authors have shown in their data that there is hardly any covariation of this variable with subjective reports. So please give example of other interoceptive tests. If there are none the limitations of this approach should be addressed. I think one cannot conclude from the ability to count ones heartbeats to general interoceptive abilities.

Response: We fully agree with the Reviewer's assessment regarding the general challenges in assessing interoceptive abilities and the limited possibilities of the current experimental approach focusing specifically on cardiac interoception. We have therefore added the following statement [Line 98-109]:

- *Part of this dissonance may be related to the measurements used to assess interoception. Most of the abovementioned studies operationalized interoception either as individual performance on cardiac-related perception tasks or as questionnaire scores, which have low correspondence with each other (Desmedt et al., 2022a). Furthermore, although other experimental approaches exist (e.g., tasks of gastrointestinal perception, van Dyck et al., 2016, of respiratory perception, Harver et al., 1993, or of the perception of spontaneous fluctuations in electrodermal activity, Krautwurst et al., 2016), the relationship between interoceptive abilities and psychopathology has mostly been tested with cardiac-related tasks, limiting the generalizability to other domains (Desmedt et al., 2022a, 2022b, Brener & Ring, 2016). Developing new tools and taxonomies that help homogenize measurements of interoception would thus help improve our understanding of the relationship between interoception and psychopathology.*

Comment 9: Line 112-114 Issue (3) is difficult to understand, please explain in a different or more detailed way or give an example.

Response: We agree that our elaboration of the term "interoceptive awareness" in its present form was difficult to understand. We have therefore further elaborated the statement as follows:

- *Within interoceptive-related tasks, interoceptive awareness is typically assessed by calculating the correspondence between objective performance (i.e., interoceptive accuracy), and the beliefs about performance (i.e., interoceptive sensibility), with higher correspondence indicating higher interoceptive awareness (Garfinkel et al., 2015). [Line 119-123]*

Comment 10: Line 140 – 144 Here it would be good to add a sentence with likely interpretations for these findings.

Response: We have added the following sentence, which refers to Pace-Schott et al. (2019):

- *(...) indicating that subjective interoceptive accuracy (i.e., a precise representation of physiological changes) and attention (i.e., a heightened attentiveness towards physiological changes), are independent traits with seemingly opposing associations with self-reported psychological traits (e.g., Pace-Schott et al., 2019). [Line 153-157]*

Comment 11: Line 165-168. It would be helpful to list at one instance in the ms. the psychometric properties of the English IAS. This might be here or in the Results or Discussion section.

Response: This information has been added in the methods section [Line 268-272]:

- *The original English IAS Version showed good psychometrical properties, for example, good to excellent internal consistency, $.88 < \alpha < .90$ (p. 119 ff.) and good test-retest reliability, $r(115) = .75, p < .001$ (p. 120 ff.). Furthermore, the authors provided evidence for convergent*

and divergent validity of the construct of subjective interoceptive accuracy (Murphy et al., 2020).

Further, we added the goodness of fit values in the discussion:

- *(e.g., Murphy et al., 2020, RMSEA = .085 90 % CI [.077, .093], CFI = .806, TLI = .784, p.127) [Line 715-716]*

Comment 12: Line 257ff please always mention if scales are validated in German (or not).

Response: This information has been now added see [Lines 274, 283, 296, 302, 308, 316, 321-324, 331-332, 338, 344, 348]

In the case of the BPQ-(V)SF, we clarified that we used the existing German version of Porges (1993) and Cabrera et al. (2017), but changed some items because they were oddly worded in German.

Therefore, we have clarified [Lines 299-301] as follows:

- *Some items of the existing German version of the BPQ-VSF and BPQ-SF that appeared oddly phrased, were reworded (the translation procedure of such items was similar to the translation of the IAS).*

Comment 13: Line 270 -273. This sentence is difficult to understand. Please try to find different wordings and/or explain the constructs (supra- vs. subdiaphragmatic; autonomic dysregulation vs. ANS reactivity).

Response: We thank the Reviewer for pointing this out. We now described the construct in detail: [Lines 287-292]:

- *(...) supradiaphragmatic reactivity (i.e., the autonomically-innervated response of organs above the diaphragm, 15 items) and subdiaphragmatic reactivity (i.e., the autonomically-innervated gastrointestinal organs, 6 items; one item referring to the feeling of likeliness to vomit is used in both the supra-and subdiaphragmatic scales), assess the construct of subjectively perceived autonomic nervous system reactivity related to difficulties in the coordination of bodily functions as well as symptoms of stress and autonomic dysregulation.*

Comment 14: Line 337-349ff. It is not clear whether HCT and SCT are two different task or part of the same task. Is the SCT seen as a test of interoceptive ability? Or as a control condition? Please carefully review that paragraph and separate the tests if possible (also with respect to the references).

Response: We apologize for any confusion caused by the lack of clarity in that paragraph. To clarify, the SCT was utilized as a control task in our study, rather than being considered a test of interoceptive ability [Lines 365-368]:

- *To ensure that the interoceptive accuracy scores extracted from HCT did not reflect any counting strategy (e.g., estimation of the heartbeats based on the time passed) a control, time estimation task was administered (second counting task, SCT; Desmedt et al., 2020; Murphy et al., 2018). In the SCT, participants are instructed to count the seconds that pass in a specific time interval.*

We have also deleted the mention of the SCT in the heading of the relevant paragraph to avoid any confusion that this might cause [Line 360].

For further clarification, we also added analysis on interceptive accuracy, using the SCT as a controlled variable in supplementary material S11)

Comment 15: Line 418 second word: this should read 'criterion' instead of 'predictor'. Why was the regression done backward in time rather than a forward prediction?

Response: Amended.

Comment 16: Line 427 to 436. The description of these variables is not detailed enough to understand/replicate them. It is suggested to move this part to the description of the respective tasks.

Response: We apologize for the lack of clarification. We now moved the description of the corresponding tasks to Lines 396-406, where the tasks are also described. We consider that the description of the variables together with the tasks can further increase comprehensibility.

Comment 17: Table 1: The fit indices of the CFA shows an unsatisfying fit in all samples. This challenges the one factor solution and the suitability of the German translation. It is suggested to take this issue up in more detail in the discussion (currently it is only shortly mentioned in limitations). Please also mention if the fit is better or worse than in the English version. How can this result be explained? What are the consequences for the application of the German IAS and for future research? Should the scale be modified or shortened to arrive at a better fit?

Response: We thank the Reviewer for pointing this out. We agree that the low fitting values invite to explore other factor solutions in future studies. We now discussed this point in the discussion section [Lines 718-724]

- *For instance, in a recent study, Brand et al. (2022) showed that allowing certain items to correlate (i.e., restrict a model for correlated residuals as a non-theoretical, more data-driven approach) improves the goodness of fit for the one-factor IAS solution (RMSEA = .072 90 % CI [.067, .078], CFI = .958, TLI = .952, p.169). This may indicate that the questionnaire may benefit from grouping different sensations under more overarching terms, as it may be that some sensations are highly correlated and/or hard to perceive (see also Murphy et al., 2020).*

We would like to mention that the cited study by Brand et al. uses a strongly data-driven approach in a different context (latent modeling). We believe that this is not an appropriate approach for the validation of the questionnaire, but it is worth mentioning it as a potential future venue here.

Comment 18: Line 481. It is not clear why the retest reliability of just this test is reported here. Please provide some context information.

Response: We provided additional information in this regard:

- *Similar to previous studies on the validation of the IAS (e.g., Murphy et al., 2020), test-retest reliability was calculated for the Body Awareness scale of the BPQ-SF, which in contrast to Murphy and colleagues results, $r(115) = .68, p < .001$ (p. 120), showed a poor test-retest reliability, $r_p(57) = .47, r_s(57) = .45, ICC = .46$. [Line 510-513]*

Comment 19: Line 600. I don't share the interpretation that the German word 'Bewusstsein' as used e.g. in 'Körperbewusstsein' (body awareness) is closer to accuracy than attention. Thus, I think this statement is debatable.

Response: We agree that this interpretation is rather debatable. Therefore, we now deleted it from the main manuscript [Former Line 600-606].

Comment 20: Line 616-618. It is not clear how this lack in covariation "provide(s) support for existing theoretical accounts of interoception...". The finding is obviously contrary to the hypotheses of the theoretical frame the authors use. This warrants a more refined discussion and interpretation, since such results challenge the whole approach of the IAS...

Response: Although certain evidence has been provided for the validity of subjective interoceptive

accuracy (e.g. the relationship with confidence ratings), we have decided to make the following more critical statement:

- *Contrary to what was expected, in the present study no evidence for a significant association between objective interoceptive accuracy, as extracted from the HDT and HCT, and IAS scores was found. [Line 642-644]*
- *However, subjective interoceptive accuracy as indexed by confidence ratings of the HCT (i.e., interoceptive sensibility) was positively associated with IAS scores. [Line 648-649]*
- *It must also be noted that objective measures of interoceptive accuracy, extracted from the HCT and HDT were unrelated. Although these results were somewhat unexpected, they are not at odds with existing data, as indicated in a recent meta-analysis where only a small association was found between the objective scores of both measures (Hickman et al., 2020). These findings thus suggest that scores from both tasks may tap into somewhat different aspects of interoception due to differing tasks demands (Hickman et al., 2020). Although the current findings may provide initial evidence for the construct of subjective interoceptive accuracy, future research on the taxonomy of interoception and the associated objective and subjective correlates is warranted. [Line 649-657]*

Comment 21: Line 634 – 636 This interpretation implicitly assumes that subjective introspective accuracy and objective accuracy are highly correlated/the same. However, the authors' data above have shown that this is not the case. Thus, it cannot be said that these findings are "in line with" Smith et al. 2021, 2022.

Response: We thank you for pointing this out. We have amended the statement [Lines 670–672]:

- *Further evidence for an association between interoception and psychopathology comes from recent studies showing deficits in cardiac interoceptive accuracy across clinical patients.*

Also, we added the following statement [Lines 676-681]:

- *Despite observing a similar negative association between interoception and psychopathology symptoms, ours and Smith et al.'s (2021) results are based on different measures of interoception which might be unrelated (see above). Future studies should thus focus on identifying the overlapping mechanisms underlying the dimensions of subjective and objective measures of interoception that may predict psychopathological symptom burden.*

Comment 22: Line 679-680 This statement is in sharp contrast to the earlier interpretation that a slightly different understanding of the word "Bewusstsein" in German will result in substantial differences.

Response: We totally agree with the Reviewer on the contradiction (see also comment 19 from Reviewer 2). We have now reworded the sentence as:

- *Although there were subtle differences in wording (e.g., regarding formality, or the use of different but synonym words of "accuracy" [genau vs akkurat]; see section S1 in the supplementary material) we found no evidence that these differences impacted the psychometric properties. [Line 727-730]*

Comment 23: Line 687 This statement can be debated. In my view there is some support but also some problems (e.g. lack of correlation with objective measures). Also one half of the information with respect to the 2x2 model is missing, since only accuracy is addressed here. Thus, this statement seems to be a bit too bold in the light of the data reported.

Response: We thank Reviewer 2 for this comment, we agree on it and therefore reformulated the last paragraph [Lines 739-743]:

-
- *Our findings add further information to the yet very heterogenous empirical evidence on interoceptive abilities and may help understand and refine common theoretical frameworks like the 2x2 factorial model of interoception. Furthermore, the present study emphasizes the need to distinguish between different constructs of interoception in relation to psychopathological symptom burden.*

13th Jul 23

Dear Mr. Brand,

Your manuscript titled "Bridging the Gap between Interoception and Mental Health: The German Validation of the Interoceptive Accuracy Scale (IAS) and its Relation to Psychopathological Symptom Burden" has now been seen by our reviewers, whose comments appear below. In light of their advice I am delighted to say that we are happy, in principle, to publish a suitably revised version in Communications Psychology under the open access CC BY license (Creative Commons Attribution v4.0 International License).

We therefore invite you to revise your paper one last time to address the remaining concerns of our reviewers and a list of editorial requests. At the same time we ask that you edit your manuscript to comply with our format requirements and to maximise the accessibility and therefore the impact of your work.

EDITORIAL REQUESTS:

If you have any questions or concerns about any of our requests, please do not hesitate to contact me. I will be out of the office from July 15-30. During this time please contact Marike Schiffer: marike.schiffer@nature.com.

SUBMISSION INFORMATION:

OPEN ACCESS:

Communications Psychology is a fully open access journal. Articles are made freely accessible on publication under a [CC BY](http://creativecommons.org/licenses/by/4.0) license (Creative Commons Attribution 4.0 International License). This license allows maximum dissemination and re-use of open access materials and is preferred by many research funding bodies.

For further information about article processing charges, open access funding, and advice and support from Nature Research, please visit <https://www.nature.com/commspsychol/article-processing-charges>

At acceptance, you will be provided with instructions for completing this CC BY license on behalf of

all authors. This grants us the necessary permissions to publish your paper. Additionally, you will be asked to declare that all required third party permissions have been obtained, and to provide billing information in order to pay the article-processing charge (APC).

* **DATA AVAILABILITY:**

[link redacted]

Best regards,

Jennifer Bellingtier

Jennifer Bellingtier, PhD
Senior Editor
Communications Psychology

REVIEWERS' EXPERTISE:

Reviewer #1 Interoception, Mental health
Reviewer #2 Mental health, psychometrics

REVIEWERS' COMMENTS:

Reviewer #1 (Remarks to the Author):

I have read the rebuttal and the revised version of the manuscript and am satisfied with the changes that the authors have made.

The main issues such as correcting for multiple comparisons, clarifying methodological and analytical details, and providing further information regarding the various IAS translations have been addressed and included in the manuscript and/or in the supplementary material.

The authors also managed to tidy up their story concerning the methodological similarities and differences between centres, e.g., by offering more systematic explanations and additional analyses. Overall, this adds to the credibility of the paper and justifies the validation of the IAS using the different data sets.

The paper is interesting to read, and the results including their interpretation contribute to the literature on interoceptive measurements. Specifically, the validation of the German IAS, and the final German version made available in S2 will help to improve assessing IAS through the use of a consistent measure in the German speaking community.

Reviewer #2 (no further remarks to the author)